**communications**

**biology**

# Joint control of visually guided actions involves concordant increases in behavioural and neural coupling

David R. Painter [1,2,3✉], Jeffrey J. Kim [1,4], Angela I. Renton [1] & Jason B. Mattingley [1,4,5]

It is often necessary for individuals to coordinate their actions with others. In the real world, joint actions rely on the direct observation of co-actors and rhythmic cues. But how are joint actions coordinated when such cues are unavailable? To address this question, we recorded brain activity while pairs of participants guided a cursor to a target either individually (solo control) or together with a partner (joint control) from whom they were physically and visibly separated. Behavioural patterns revealed that joint action involved real-time coordination between co-actors and improved accuracy for the lower performing co-actor. Concurrent neural recordings and eye tracking revealed that joint control affected cognitive processing across multiple stages. Joint control involved increases in both behavioural and neural coupling – both quantified as interpersonal correlations – peaking at action completion. Correspondingly, a neural offset response acted as a mechanism for and marker of interpersonal neural coupling, underpinning successful joint actions.

[1] The University of Queensland, Queensland Brain Institute, St Lucia, Australia. [2] Hopkins Centre, Menzies Health Institute Queensland, Griffith University, Gold Coast, Queensland, Australia. [3] Menzies Health Institute Queensland, Griffith University, Gold Coast, Queensland, Australia. [4] The University of Queensland, School of Psychology, St Lucia, Australia. [5] Canadian Institute for Advanced Research (CIFAR), Toronto, Canada. ✉email: david.ross.painter@gmail.com

Experimental investigations of human behaviour have traditionally focused on the individual in isolation from their social context. However, increasing evidence suggests that behaviour and cognition may depend on interactions between individuals[1–10]. This is particularly evident in joint actions, such as synchronised rowing[11,12] and dancing[13,14], which occur when two or more individuals coordinate their actions in space and time toward a common goal. Laboratory investigations demonstrate that such joint actions depend on the abilities to share goal representations, predict each other's actions, and integrate these predictions with one's own actions[15–20]. This implies that joint actions involve additional cognitive load, and thus might result in a behavioural cost relative to the same actions performed individually[21]. However, on object-oriented tasks affording co-actors the same action possibilities (e.g., shared horizontal and vertical movement), known as redundant object control, groups may outperform individuals[17,22–27].

A variety of high-level factors may contribute to the performance benefits of redundant control, including the presence of sensory cues indicating the co-actor's actions, the information carried by those cues, and the ability to select actions consistent with individual preferences[28]. At a lower level, it is less clear which cognitive mechanisms might contribute. In general, joint actions may involve behavioural and neural coupling, which manifest as interpersonal correlations in behaviour and cognition, effectively linking the perceptual and motor systems of co-actors through shared sensory input and common motor execution[7,23,25,29–33]. Joint actions also involve alterations in selective attention, namely, the ability to prioritise task-relevant objects and suppress distracting stimuli, since co-actors jointly attend to common sensory inputs[15,16,34–36].

To elucidate redundant control, we sought to identify the most impactful cognitive processing stages and mechanisms. Previous human studies have tended to focus on continuous actions without discrete start and endpoints[17,37], thereby obscuring distinct processing stages[38,39]. Additionally, previous human studies have tended to measure behavioural performance in isolation, without considering how concurrent brain activity might reveal latent processes associated with task performance[40,41] (although we note that recent technological advances have made neural recordings during joint action increasingly tractable[20,42]). Here we addressed these issues by having pairs of participants perform discrete, object-oriented actions while we recorded eye movements and brain activity using electroencephalography (EEG)[43,44]. Across trials, cursor control was either shared redundantly between individuals within the pair (joint control) or was ceded to each individual independently (solo control). This allowed us to segment cognitive processing into stages of action preparation, initiation, execution, and completion. Additionally, our multimodal approach allowed us to assess potential contributions of interpersonal neural coupling and visual selective attention. We compared joint control to the same actions performed individually, allowing us to closely match visual inputs and motor outputs, thereby isolating the pure effects of joint action on behaviour and brain activity[30].

Using this approach, the results provided a concrete mechanism for interpersonal neural coupling that excludes confounding factors and links neural coupling specifically to real-time coordination between co-actors and successful and precise joint actions. The results show a direct brain-behaviour link underpinning neural coupling arising from coincident motor offsets between co-actors during joint control. The analyses identify a phase-locked electrophysiological component that underlies neural coupling under joint control.

## Results

**Task overview.** Pairs of participants ($N = 40$ individuals divided into 20 pairs) were seated adjacent to each other and were separated by a partition so they could not observe their partner or their partner's display (Fig. 1a). Participants used video game controllers to guide cursors (red or green dots and surrounding annuli) to a target (black annulus) at the periphery of the display, while ignoring a distractor positioned in the opposite hemifield (Fig. 1b). Actions were considered complete/accurate at the time when the cursor first entered and remained within the target. On randomly interleaved trials, an action control cue changed colour (to green or red), informing participants with 100% validity that they were about to perform a solo or joint action. The action control cue contingency was made explicit to participants, and cue colours were counterbalanced across participant pairs (Fig. 1c). Target locations (Fig. 1d) were spatially cued with 100% validity by a small central arrow.

On solo control trials, participants had complete control of their own cursor, viewed only on their own display, appearing concurrently on each trial for each co-actor of the pair. On joint control trials, participants had joint control of a common cursor and viewed identical visual displays (Supplementary Movie 1). Joint cursor movement was calculated as the mean of the inputs from the two co-actors, with the requirement that both co-actors had to exert force on the controller concurrently to elicit cursor movement. Participants were instructed to emphasise both the speed and precision of their solo and joint control actions and were not permitted to communicate with each other. Participants were told that they would share control on joint trials and that, to be successful, they would have to work cooperatively with their partner. No haptic feedback regarding the partner's actions was available to participants via the game controllers.

**Analyses of cognitive processing stages and cognitive mechanisms.** To measure the effects of action preparation on latent processing, the action control cue flickered at 7 Hz, evoking a frequency-tagged steady-state visual evoked potential (SSVEP) that reflected pre-action stimulus-driven neural activity[45–47]. Action initiation was measured using action onset time. Action execution was measured using movement time, accuracy, and cursor trajectories. Overt spatial attention was measured using eye gaze position, similarity in gaze position between co-actors, and the relative timing of cursor and gaze movements. To measure the electrophysiological effects of spatial selective attention, circular textured patches at the target and distractor locations flickered at distinct frequencies (17 and 19 Hz), thus evoking distinct SSVEPs. Interpersonal coupling was measured using interpersonal correlations in controller displacement, action onsets, action offsets, and EEG recordings of brain activity.

## Overall task performance

*Overview.* The principal goal of the behavioural analyses was to establish whether any differences between solo and joint control were attributable to real-time coordination between the co-actors or instead to signal averaging (i.e., the co-actors in effect performing joint actions independently)[48,49]. We assessed this by contrasting solo and joint cursors on both solo and joint trials. On joint trials, solo cursor trajectories were calculated during the experiment but were invisible to participants. Similarly, we contrasted behavioural performance for joint cursors between solo and joint trials. On solo trials, joint cursor trajectories were calculated but were again invisible. We refer to the visible cursors experienced directly by participants, and the invisible cursors that were always present but hidden from view. If behavioural performance was attributable to signal averaging, we would expect identical performance for visible and invisible cursors as participants should perform identically without regard to their co-actor. If, by contrast, behaviour reflects real-time coordination,

## Solo and Joint Control Visuomotor Action Tasks

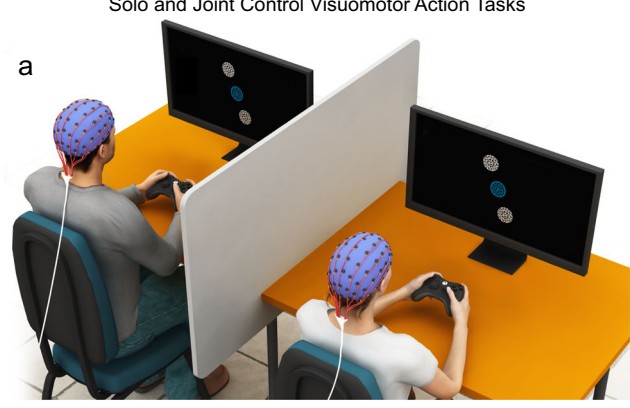

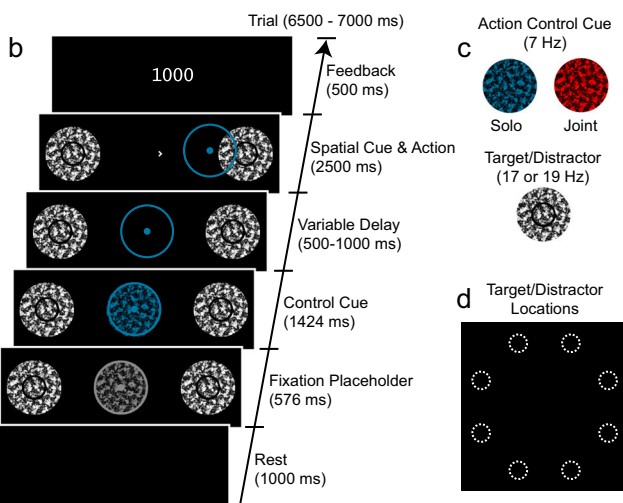

**Fig. 1 Solo and joint control visuomotor action tasks with multi-modal recording (*n* = 20 pairs). a** Co-actors viewed their own display and used video game controllers to move the cursor to the spatially cued peripheral target, while ignoring a distractor positioned at 180° in the opposite hemifield. The artwork was provided by our colleague Dr David Lloyd. **b** Example trial sequence shown for a segment of the display (enlarged and rotated here for clarity). Participants guided the coloured cursor (a dot and surrounding annulus) to the spatially cued target location (black annulus within the circular textured patch). The position of the cued target within the peripheral array was indicated by a small, central arrowhead (in white), which served as the action onset cue. The green- or red-coloured action control cue and cursor indicated solo or joint control. Note that the green cursor and control cue is depicted as blue in this and subsequent figures for the benefit of colour-blind readers. Numeric feedback for correct responses reflected reaction time (RT) plus movement time (MT). If participants performed the task incorrectly, feedback indicated that the action was "too fast", "too slow" or had "missed the target" (i.e., the cursor's central dot had not entered and remained within the target annulus, indicated in black). **c** Frequency-tagged visual stimuli. The action control cue and target and distractor stimuli flickered at distinct frequencies, thereby evoking frequency-defined evoked neural oscillations, allowing the measurement of selective attention prior to and during task performance. **d** All possible target/distractor positions within the display. On each trial, only one target and one distractor were presented (with 180° separation between them).

performance should differ between visible and invisible cursors (Fig. 2a).

Additionally, the analyses aimed to contrast overall performance patterns between solo and joint control. Thus, each pair of participants was divided at the experiment level into a higher performing (HP) and lower performing (LP) co-actor separately

for each performance metric: accuracy, reaction time (RT), and movement time (MT). This allowed consistency between these metrics, as the HP solo actor cannot be identified with precision at the single-trial level on the dichotomous variable of accuracy (with levels of correct and error).

For visible and invisible cursors, values on each metric were computed in real-time during the experiment using the same game logic. Actions were considered correct if RTs were neither too fast (<200 ms) nor too slow (>800 ms). RTs <200 ms were treated as incorrect during the experiment since they were assumed likely to reflect anticipations rather than responses to the spatial cue to move. Note that the 200 ms cutoff is below the average RT for human observers on simple detection tasks[50]. For visible cursors, anticipations (%) were infrequent overall and were significantly more common under solo control ($M = 3.66$, $SD = 3.25$) compared with joint control ($M = 1.92$, $SD = 3.07$; $t_{39} = 4.15$, $p = 1.73$e-04). MTs were defined only on correct trials, which additionally constrained correct trials as requiring that the MT not be too slow (>1500 ms) and that the cursor entered and remained within the target (for >200 ms).

*Joint control is attributable to real-time coordination rather than signal averaging.* The behavioural metrics (accuracy [%], MT [ms], RT [ms]) were submitted to two-way repeated measures ANOVAs with the factors of cursor visibility (visible, invisible) and control (LP solo, HP solo, joint). The hypothesis of real-time coordination between co-actors predicted statistically significant cursor visibility × cursor control interactions and significant main effects of cursor visibility. Consistent with real-time coordination, all the behavioural metrics showed significant interactions between cursor visibility × cursor control (Fig. 2b, accuracy: $F_{2,38} = 7.81$, $p = 0.001$, $\eta_p^2 = 0.291$; Fig. 2c, RT: $F_{2,38} = 14.96$, $p = 1.61$e-05, $\eta_p^2 = 0.441$; Fig. 2d, MT: $F_{2,38} = 7.30$, $p = 0.002$, $\eta_p^2 = 0.277$). Additionally, the main effects of cursor visibility were significant for accuracy ($F_{1,19} = 780.45$, $p = 6.81$e-17, $\eta_p^2 = 0.976$) and MT ($F_{1,19} = 16.89$, $p = 5.96$e-04, $\eta_p^2 = 0.471$) but not for RT ($F_{1,19} = 0.48$, $p = 0.496$, $\eta_p^2 = 0.025$). Notably, accuracy was statistically and substantially higher for visible ($M = 74.36$, $SD = 13.81$) compared with invisible cursors ($M = 11.74$, $SD = 9.18$). These results show that cursor visibility was necessary for successful task performance and that participants changed their performance across solo and joint control trials, consistent with real-time coordination under joint control. Note that all source data underlying the graphs presented in the main figures is available (Supplementary Data 1).

*Compared with LP solo control, joint control improved accuracy and MT but slowed RT.* The significant cursor visibility × cursor control interactions were followed up for visible cursors with the simple effects of control (LP solo, HP solo, joint) – all these effects were significant (accuracy: $F_{2,38} = 20.60$, $p = 8.72$e-07, $\eta_p^2 = 0.520$; RT: $F_{2,38} = 54.84$, $p = 6.29$e-12, $\eta_p^2 = 0.743$; MT: $F_{2,38} = 49.85$, $p = 2.38$e-11, $\eta_p^2 = 0.724$). Follow-up tests showed that accuracy was significantly higher for joint control ($M = 79.21$, $SD = 13.38$) compared with LP solo control ($M = 64.56$, $SD = 12.23$; $t_{19} = 6.22$, $p = 5.63$e-06) but did not differ significantly between joint and HP solo control ($M = 79.31$, $SD = 10.59$; $t_{19} = 0.05$, $p = 0.958$). Similarly, MT was significantly faster for joint control ($M = 641.44$, $SD = 119.97$) compared with LP solo control ($M = 781.63$, $SD = 126.47$; $t_{19} = -5.16$, $p = 5.52$e-05) but was significantly slower for joint control compared with HP solo control ($M = 586.57$, $SD = 108.99$; $t_{19} = -8.76$, $p = 4.21$e-08). By contrast, RT was significantly slower for joint control ($M = 505.13$, $SD = 54.16$) compared with both LP solo control ($M = 482.83$, $SD = 57.84$; $t_{19} = -7.72$, $p = 2.86$e-07) and HP solo control ($M = 419.99$, $SD = 45.71$; $t_{19} = -9.28$, $p = 1.74$e-08). Thus, overall, real-time coordination benefited the LP solo actor on accuracy and MT but at a cost to RT.

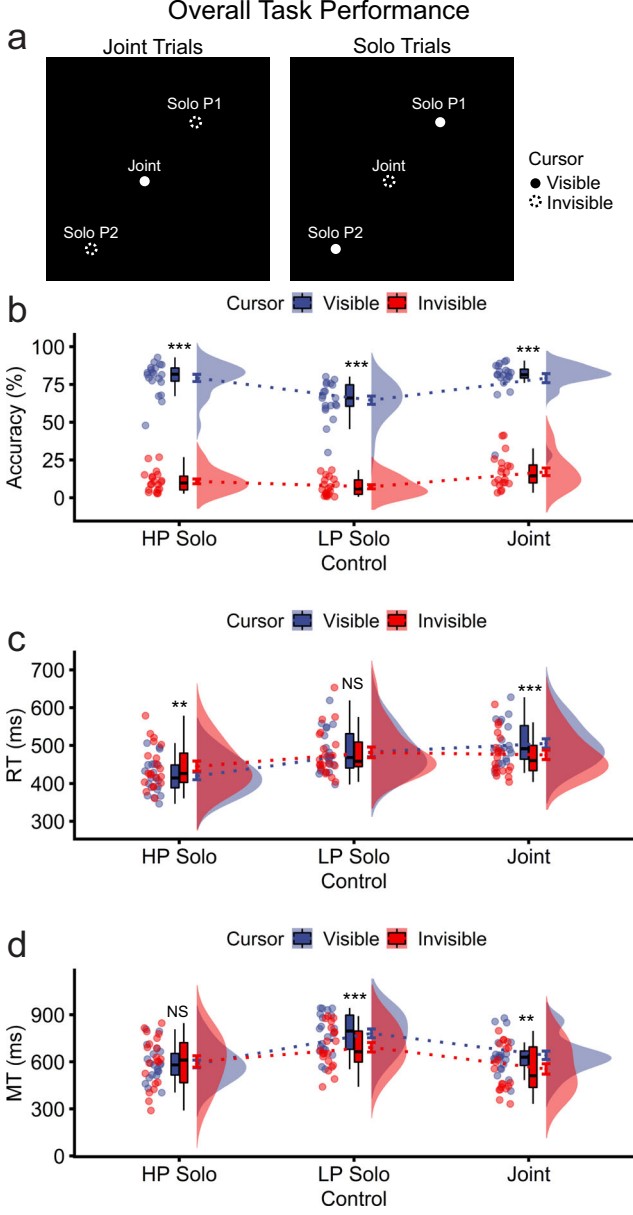

**Fig. 2 Overall behavioural performance for solo and joint control (n = 20 pairs).** Solo performance metrics were sorted by the high performing (HP) and low performing (LP) co-actor at the experiment level, separately for each metric. **a** On joint action trials, a joint cursor was visible to participants, and trajectories of invisible solo cursors of Participant 1 (P1) and Participant 2 (P2) were not displayed. On solo trials, the solo P1 and P2 cursors were individually visible to the corresponding participant (P1 or P2), and the joint cursor was invisible. **b** Raincloud plots[72] of accuracy in behavioural performance. For this and subsequent raincloud figures, dots reflect individual participants (for solo control) or participant pairs (for joint control), boxplots indicate the median, the 25th/75th percentiles (at the upper/lower hinges) and 1.5 × interquartile range from the hinges (whiskers). The violin plots show the kernel density estimation, and the error bars (here coloured blue and red) reflect $M \pm SE$ $(\frac{SD}{\sqrt{N}})$. Cursor movements were considered errors if they were too fast, too slow or had missed the target. **c** RT for correct trials, calculated as time elapsed between spatial cue onset and action initiation. **d** MT for correct trials, calculated as time elapsed from action onset to action completion. For this and subsequent figures: NS $p > 0.050$, *$p < 0.050$, **$p < 0.010$, ***$p < 0.001$. In this figure, statistical significance annotations reflect the simple effects of cursor visibility at each level of cursor control, emphasising that joint control was attributable to real-time coordination between co-actors rather than signal averaging of actors performing independent solo tasks.

control and HP solo control ($M = 0.26$, $SD = 0.06$; $t_{19} = -0.98$, $p = 0.339$; Fig. 3b).

We repeated this analysis, examining endpoint displacement. As depicted in Fig. 3c, endpoint displacement (°) appeared smaller for joint compared with LP solo control. This was confirmed statistically, with all ANOVA-level effects significant ($Fs \geq 51.41$, $ps \leq 1.55e-11$, $\eta_p^2 s \geq 0.730$). The follow-up tests for visible cursors were all significant (control effect: $F_{2,38} = 14.53$, $p = 2.06e-05$, $\eta_p^2 = 0.443$), indicating smaller endpoint displacement for joint control ($M = 0.62$, $SD = 0.22$) compared with LP solo control ($M = 1.07$, $SD = 0.79$; $t_{19} = 3.10$, $p = 0.006$). Endpoint displacement was also significantly smaller for HP solo ($M = 0.37$, $SD = 0.04$) compared with joint control ($t_{19} = -5.74$, $p = 1.58e-05$; Fig. 3d). Overall, these results show that joint control improved trajectories for the LP actor, with performance equivalent to or poorer than the HP actor. Thus, on the best-case scenario metrics, namely curvature and overall accuracy, the current results support race model accounts of joint action, with performance accounted for by statistical facilitation – picking the better of the two input signals[48,49] – in this case, the motoric signals provided by the HP actor.

### Eye gaze
*Joint control reduced inter-gaze distance.* Four participants from three participant pairs were excluded for missing or inaccurate eye tracking data. During the pre-action epoch of each trial, participants maintained central fixation and attention on the control and spatial cues (Fig. 4a). During the action epoch, participants shifted their gaze to one of the eight peripheral target locations (Fig. 4b). We examined inter-gaze distance, calculated as the Euclidean distance between the co-actors' gaze positions, quantifying the extent to which the co-actors looked at the same location on their display as the trial unfolded. Inter-gaze distance (°) was submitted to a two-way repeated measures ANOVA with the factors of control (joint, solo) and epoch (pre-action, action). All effects were significant (control: $F_{1,16} = 8.16$, $p = 0.011$, $\eta_p^2 = .338$; epoch: $F_{1,16} = 48.19$, $p = 3.32e-06$, $\eta_p^2 = 0.751$; control × epoch: $F_{1,19} = 15.40$, $p = 0.001$, $\eta_p^2 = 0.490$). Follow-up tests indicated that inter-gaze distance during the pre-action epoch did not differ

*Joint control reduced trajectory curvature and increased endpoint precision for joint control compared with LP solo control.* To further understand how joint control affects action execution, cursor trajectories were plotted by cursor visibility and cursor control. As depicted in Fig. 3a, visible trajectories appeared straighter for joint control relative to LP solo control. Trajectory curvature (°) was analysed via a two-way ANOVA with the factors of cursor visibility (visible, invisible) and cursor control (LP solo, HP solo, joint). All ANOVA-level effects were significant, and importantly, consistent with real-time coordination. The cursor visibility × cursor control interaction was significant ($F_{2,38} = 61.20$, $p = 1.31e-12$, $\eta_p^2 = 0.763$), as was the main effect of cursor visibility ($F_{1,19} = 43.99$, $p = 2.41e-06$, $\eta_p^2 = 0.698$). Follow-up tests for visible cursors showed a significant simple effect of cursor visibility ($F_{2,38} = 71.39$, $p = 1.35e-13$, $\eta_p^2 = 0.79$), with trajectory curvature significantly smaller for joint control ($M = 0.28$, $SD = 0.11$) compared with LP solo control ($M = 0.47$, $SD = 0.09$; $t_{19} = = 8.28$, $p = 9.99e-08$). Curvature was statistically equivalent for joint

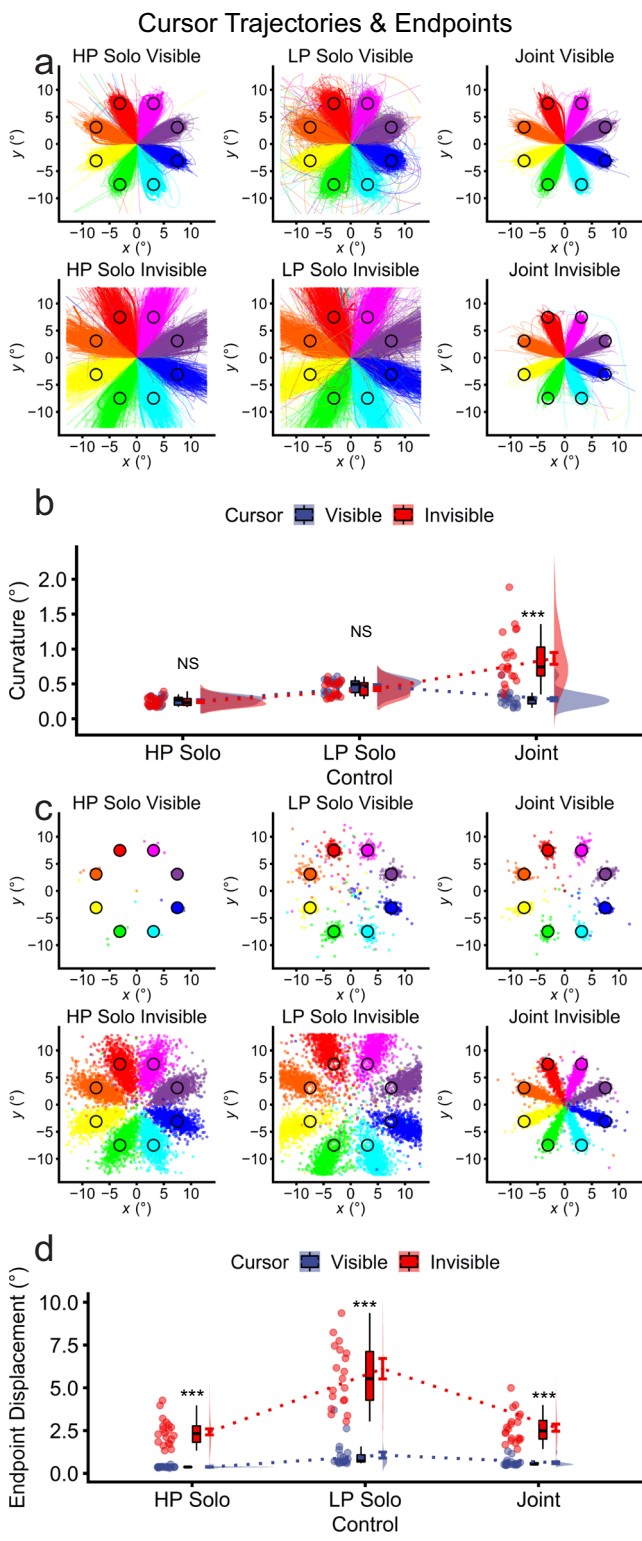

**Cursor Trajectories & Endpoints**

**Fig. 3 Cursor trajectories for the solo and joint control tasks by cursor control (HP solo, LP solo, joint) and cursor visibility (visible, invisible; $n = 20$ pairs).** LP and HP solo co-actors were determined at the single-trial level separately for visible and invisible cursors. **a** Trajectories for all participants and trials, with LP and HP solo co-actors determined by sorting the mean curvature, calculated across trial frames. Curvature was calculated as the hypotenuse of x and y curvature components. **b** Trajectory curvature averaged across frames and then trials. **c** Trajectory endpoints for all participants and trials. Solo trajectories were sorted by endpoint displacement from the target position on the final trial frame. **d** Endpoint displacement. Statistical significance annotations reflect the effects of cursor visibility at each level of cursor control, highlighting that joint control reflected real-time coordination between co-actors rather than signal averaging. For clarity, the y axis has been truncated to 10.0°, hiding an outlier for invisible LP solo control (14.4°).

solo and joint control. To illustrate this point, at 50% of action time, the distance to the target was ~2.50° for gaze and ~5.75° for the cursor (Fig. 4e, f and Supplementary Movie 2).

### Interpersonal behavioural correlations

*Joint behavioural correlations reflect real-time coordination.* Joint actions and other cooperative behaviours have been associated with interpersonal behavioural correlations[11–14,23]. To assess this, we examined correlations between co-actors in video game controller displacement time series, calculated as the Euclidean displacement from the controller home position, ranging from 0 to 100% (Fig. 5a). To investigate potential interpersonal behavioural correlations, solo and joint control actions were divided into correct and error trials, and within-pair controller displacement correlations ($r$) were submitted to a two-way ANOVA with control (solo, joint) and accuracy (correct, error) as repeated-measures factors. As depicted in Fig. 5b, there were significant effects of control ($F_{1,19} = 41.45$, $p = 3.59e-06$, $\eta_p^2 = 0.686$), accuracy ($F_{1,19} = 53.01$, $p = 6.60e-07$, $\eta_p^2 = 0.736$), and control × accuracy ($F_{1,19} = 25.59$, $p = 6.98e-05$, $\eta_p^2 = 0.574$). The main effect of control indicated that controller displacement correlations were significantly higher for joint compared with solo control. Follow-up *t*-tests showed that controller displacement correlations were significantly higher for correct compared with error trials (joint: $t_{19} = -7.45$, $p = 4.77e-07$; solo: $t_{19} = -6.05$, $p = 8.03e-06$), and this effect ($\Delta r$) was significantly larger under joint ($M = 0.20$, $SD = 0.12$) compared with solo control ($M = 0.11$, $SD = 0.08$; $t_{19} = 5.06$, $p = 6.97e-05$). These patterns provide converging evidence that joint action involved real-time coordination rather than signal averaging (Figs. 2 and 3).

To examine whether real-time coordination was greater at the beginning or end of each action, we examined Pearson's correlations between co-actors across trials of action onsets and offsets, submitting these correlations ($r$) to separate repeated-measures ANOVAs with control (joint, solo) and accuracy (error, correct) as factors. For action onsets, there was a significant main effect of accuracy ($F_{1,19} = 11.39$, $p = 0.003$, $\eta_p^2 = 0.375$), such that correlations were higher for correct ($M = 0.10$, $SD = 0.10$) than for error trials ($M = -0.03$, $SD = 0.20$; Fig. 5c). The effects of control ($F_{1,19} = 0.11$, $p = 0.739$, $\eta_p^2 = 0.006$) and control × accuracy ($F_{1,19} = 0.62$, $p = 0.443$, $\eta_p^2 = 0.031$) were non-significant, indicating that correlations in action onsets did not reliably predict action control. By contrast, for action offsets, which reflect action completion, all ANOVA-level effects were significant (control: $F_{1,19} = 95.59$, $p = 7.56e-09$, $\eta_p^2 = 0.834$; accuracy: $F_{1,19} = 16.56$, $p = 0.001$, $\eta_p^2 = 0.466$; control × accuracy: $F_{1,19} = 43.70$, $p = 2.52e-06$, $\eta_p^2 = 0.697$), indicating significantly higher correlations in action offsets for joint ($M = 0.48$, $SD = 0.22$) compared with solo trials ($M = 0.06$, $SD = 0.17$). Follow-up *t*-tests

significantly between solo ($M = 2.41$, $SD = 0.64$) and joint control ($M = 2.45$, $SD = 0.65$; $t_{16} = -0.98$, $p = 0.344$; Fig. 4c). By contrast, inter-gaze distance during the action epoch was significantly smaller for joint ($M = 3.16$, $SD = 0.65$) compared with solo control ($M = 3.44$, $SD = 0.64$; $t_{16} = 3.66$, $p = 0.002$; Fig. 4d). These results suggested that joint control involved shared attentional gaze between co-actors. Figure 4e shows the relative timing of gaze and cursor positions throughout the trial. Participants moved their gaze first to the target position and then moved the cursor for both

## Eye Gaze Patterns

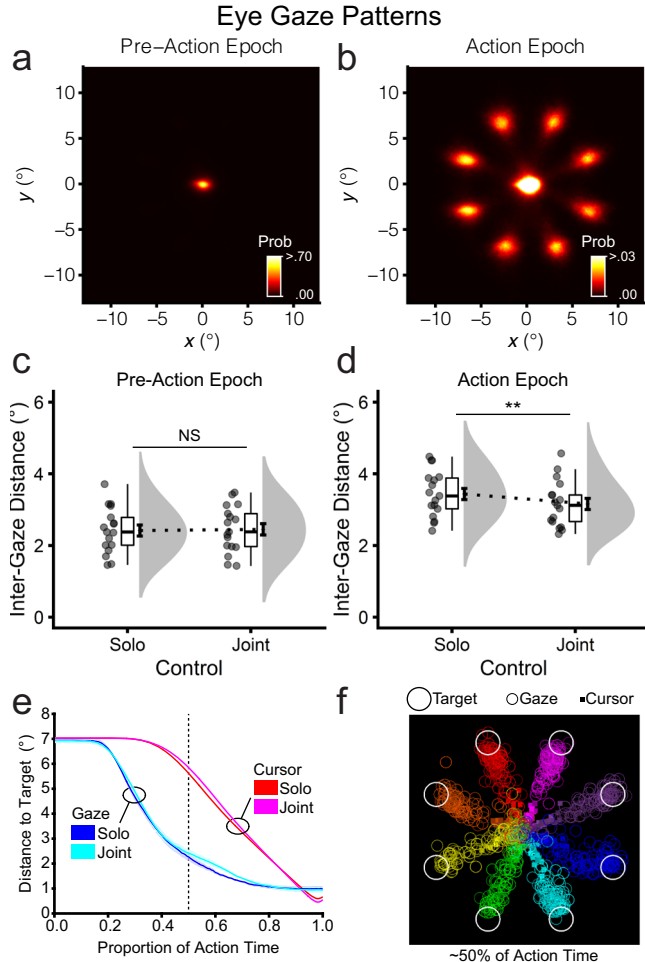

**Fig. 4 Eye gaze patterns for the solo and joint control tasks (*n* = 17 pairs). a, b** Eye gaze heat maps indicate participants' most frequent gaze positions in x and y coordinates during the pre-action epoch (−2.5–0.0 s) and action epoch (0.0–2.5 s) across all trials (N = 36 of 40 participants). **c, d** Within-pair inter-gaze distance, calculated as the absolute/directionless Euclidean displacement between gaze time series (N = 17 of 20 participant pairs). **e** Relative timing of cursor and eye gaze movement. The x-axis is normalised by the proportion of total movement time within each trial. **f** Single still frame of an animation (Supplementary Video S2) for all joint control trials of a representative participant pair at ~50% of action time, illustrating the relative timing of cursor positions (filled squares) and eye gaze positions (annuli), colour-coded by target position. At this timepoint, relative to the cursor, eye gaze had advanced closer to the target.

## Interpersonal Behavioural Correlations

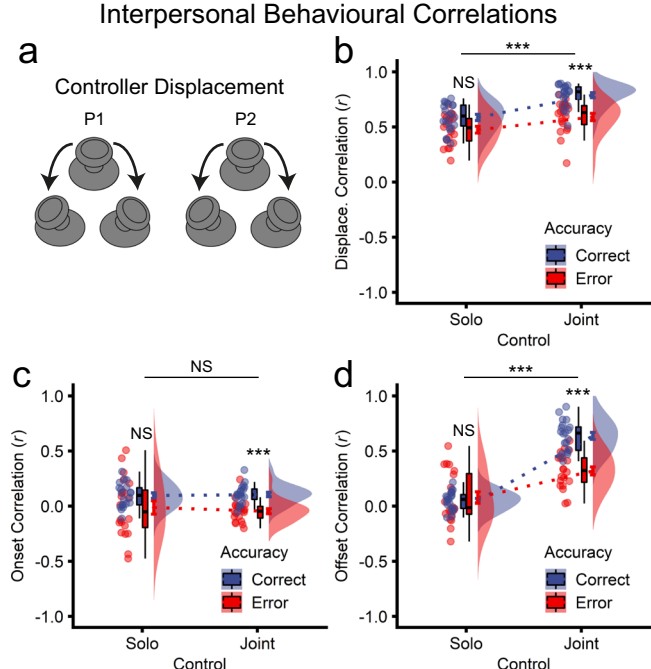

**Fig. 5 Effects of joint control on interpersonal behavioural correlations (*n* = 20 pairs). a** Controller displacement was calculated as the Euclidean displacement from the controller home position, ranging from 0 to 100%. The equation for displacement is given by the Pythagorean equation $\sqrt{x^2 + y^2}$, where x and y refer to the horizontal and vertical components, respectively. Displacement was calculated at each frame, creating a time series of displacement values. Within-pair (Pearson, zero-lag) correlations were calculated between co-actors' controller displacement time series for each trial. **b** Grand mean controller displacement correlations for solo and joint control actions, shown in different colours for correct (red) and error (blue) trials. **c** Action onset correlations. Onsets for each trial were calculated separately for each co-actor on solo and joint trials as the first frame of movement showing displacements greater than 10% of maximum. **d** Action offset correlations. Offsets were calculated separately for each co-actor as the last frame of movement showing displacements greater than 10% of maximum. Onsets and offsets, calculated for each trial, were then separately correlated between participant pairs across trials.

showed that action offset correlations for joint control were significantly higher for correct (M = 0.64, SD = 0.14) than for error trials (M = 0.32, SD = 0.16; $t_{19} = -8.08$, $p = 1.44e-07$). For solo control, correlations in action offsets did not differ significantly between correct (M = 0.05, SD = 0.09) and error trials (M = 0.08, SD = 0.23; $t_{19} = 0.55$, $p = 0.591$; Fig. 5d). Collectively, these results indicate that real-time coordination for joint control is maximal at action completion and predicts successful joint action performance.

### Stimulus-driven frequency-tagged neural activity
*Joint control increased attention to the control cue preceding action.* Before action, participants received a green- or red-coloured action control cue, which flickered at 7 Hz and indicated whether the upcoming action would involve solo or joint control

(Fig. 1b, c). Control cue SSVEP amplitudes, averaged over the 1424 ms control cue epoch, were measured as frequency-specific changes in electrical brain activity. The control cue produced a spike at 7 Hz (and harmonic multiples) in the Fast Fourier Transform (FFT) of the grand mean event-related potential (ERP), reflecting oscillatory SSVEP neural activity (Fig. 6a). To assess whether this neural activity differed in amplitude between solo and joint control, Morlet wavelet time-frequency transforms were performed on the grand mean ERP. These showed a strong band of 7 Hz activity during the control cue epoch (Fig. 6b). The subtraction contrast – joint minus solo – showed higher 7 Hz amplitudes for joint compared with solo control. To assess statistical significance at each time point, permutation simulations shuffled single-trial solo and joint control labels, and the grand mean joint-minus-solo contrast was recomputed, forming a null hypothesis distribution of no association between control (solo, joint) and 7 Hz SSVEP amplitudes (μV). This analysis, with α set at 0.001 (two-tailed) to control for multiple comparisons across timepoints, indicated that SSVEP amplitudes to the control cue were significantly higher on joint compared with solo control trials from 196 to 236 and 742 to 1384 ms post-control cue onset (Fig. 6b). The peak difference (μV) occurred at 1142 ms ($t_{max}$,

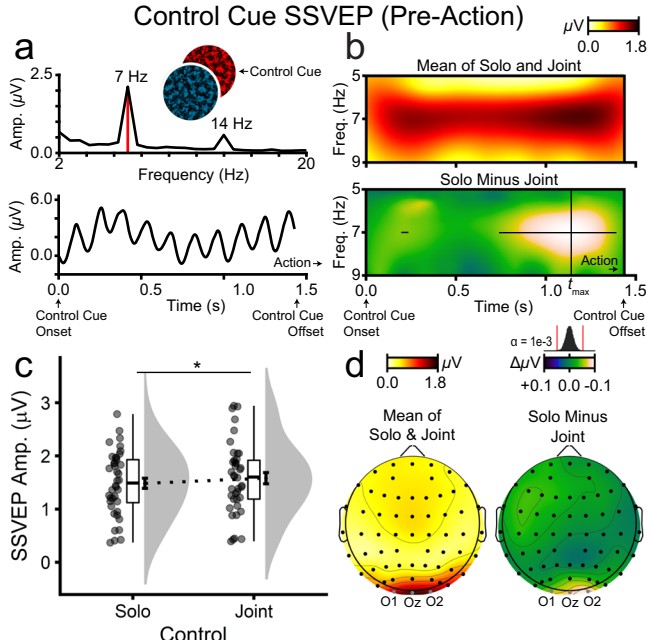

**Fig. 6 Neural activity elicited by the action control cue prior to the onset of solo and joint actions (n = 40 participants).** The action control cue (coloured green or red) indicated whether participants were to undertake a solo or joint control action upon onset of the action cue. Depicted data correspond to the pre-action trial period beginning with control cue onset and ending with control cue offset. **a** Grand mean FFT and ERP to the action control cue. The action control cue SSVEP is visible as a 7 Hz frequency-domain spike (FFT, upper panel) and time-domain oscillatory activity (ERP, lower panel). **b** Wavelet time-frequency heat maps. In the lower panel, the histogram on the bottom reflects a permutation simulation distribution with red lines indicating significance thresholds. The superimposed black lines in the lower panel indicate time points at which SSVEP amplitudes were significantly higher for joint compared with solo control actions. **c** Morlet wavelet SSVEP amplitudes at the time point of maximal deviation between solo and joint control ($t_{max}$), assessed via wavelet analysis. **d** Wavelet SSVEP amplitude topographies at $t_{max}$, revealing focused activity over occipital electrodes (O1, Oz, O2). Note that panels **b** (lower) and **d** share the same colour scale and limits.

solo vs. joint; $t_{39} = -2.65$, $p = 0.012$; Fig. 6c), with heightened responses focused at occipitoparietal electrodes of maximal response on the grand mean ERP (Fig. 6d). Notably, this enhanced neural response under joint control arose before participants received the spatial cue (see Fig. 1b for the trial timeline and note that the control cue preceded action onset). In other words, merely being cued to prepare for joint versus solo control actions triggered a reliable increase in stimulus-driven neural activity.

*Joint control did not affect attention to targets and distractors.* Prior to and during action, participants viewed two circular textured patches, each flickering at a unique frequency (17 and 19 Hz). The spatial cue designated one of these patches as the target and, by default, the other as the distractor. This display sequence enabled us to obtain a readout of the brain's response to the target and distractor during action preparation and execution. As is evident from Fig. 7a, b, which shows the frequency spectra for the target and distractor, when the target flickered at 17 Hz, there was a strong neural response at that frequency and little response to the 19 Hz distractor (Fig. 7a). Conversely, when the target flickered at 19 Hz, there was a strong response at 19 Hz and

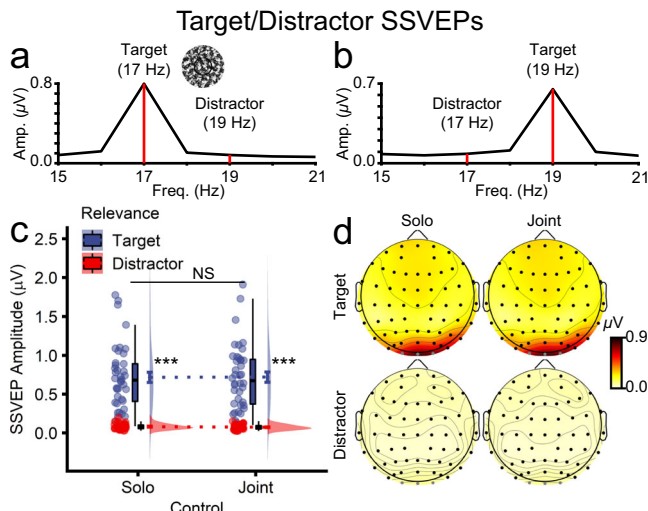

**Fig. 7 Neural activity elicited by targets and distractors during solo and joint actions, corresponding to a time window from 0.0 to 2.0 s after action cue onset (n = 40 participants).** FFT amplitudes when the target flickered at **a** 17 Hz or **b** 19 Hz. FFT SSVEP amplitudes for target and distractor stimuli shown separately for solo and joint control actions, represented as **c** raincloud plots and **d** topographies. Note that target and distractor stimuli were always presented concurrently on each trial and that their flicker frequencies were counterbalanced.

little response to the 17 Hz distractor (Fig. 7b). SSVEP amplitudes (μV) to these stimuli were submitted to a two-way repeated measures ANOVA with control (solo, joint) and stimulus relevance (target, distractor) as factors. As depicted in Fig. 7c, there was a significant main effect of stimulus relevance such that SSVEP amplitudes were significantly and substantially higher for targets ($M = 0.72$, $SD = 0.42$) than distractors ($M = 0.08$, $SD = 0.03$; $F_{1,39} = 102.80$, $p = 1.73\mathrm{e}{-12}$, $\eta_p^2 = 0.725$). The effects of control ($F_{1,39} = 0.07$, $p = 0.796$, $\eta_p^2 = 0.002$) and control × stimulus relevance ($F_{1,39} = 0.29$, $p = 0.591$, $\eta_p^2 = 0.007$) were non-significant. Figure 7d shows the SSVEP topographies for the target and distractor, separately for solo and joint control. While there was a strong neural response to the target over occipito-parietal electrodes, there was no discernible response to the distractor. This distractor suppression likely depends on participants moving their gaze to the target location (i.e., overt attention). To reiterate, however, spatial attention to the targets and distractors – as reflected by SSVEP amplitudes – did not differ between solo and joint control.

### Interpersonal neural correlations
*Joint control increases neural correlations between co-actors.* Joint actions and other cooperative behaviours have been associated with interpersonal neural correlations during active task performance[4,7,30,32,33,42,51–53]. Such correlations might arise mechanistically through shared sensory input and corresponding temporally overlapping neural processes, and theoretically might index real-time coordination between cooperating individuals. To examine interpersonal neural correlations between participants within each pair under joint and solo control, we calculated (lag-zero) Pearson's correlations ($r$) in the EEG time series between the two participants within each pair. The correlation was performed separately at each electrode using a sliding window (500 ms length in 1.56 ms increments) from the beginning to the end of the trial (−2.5 s and +3.0 s relative to the spatial cue, respectively). As depicted in Fig. 8b, c, for both solo and joint control, interpersonal neural correlations (indicated by the red-

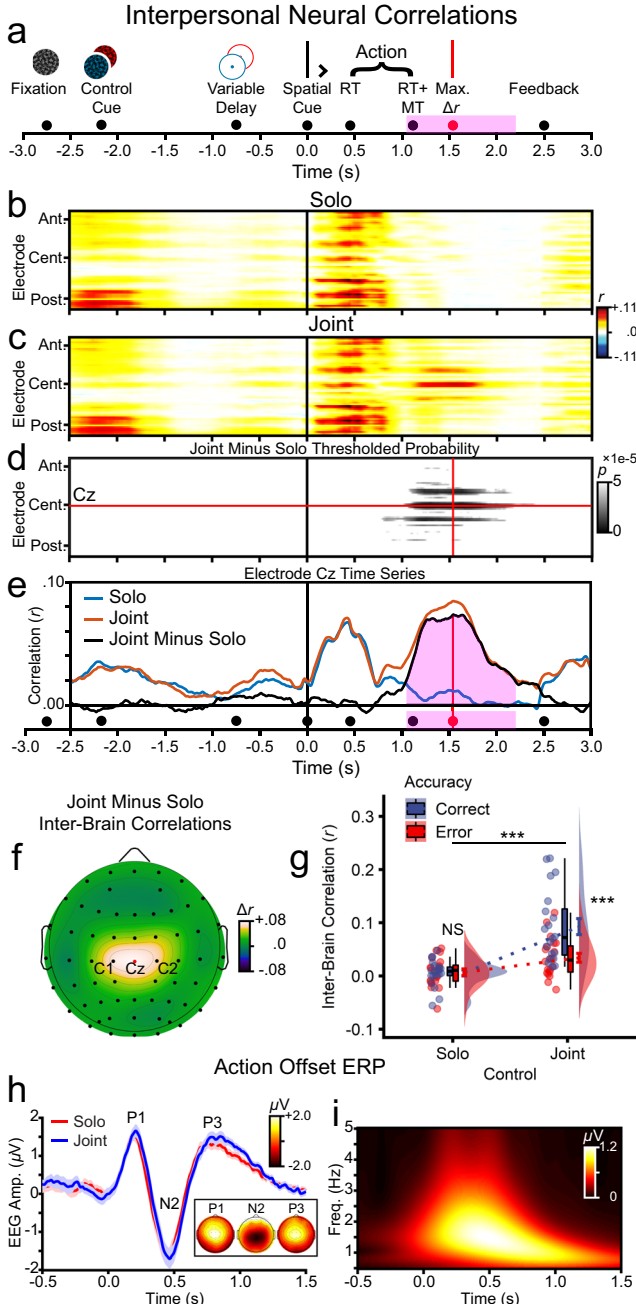

**Fig. 8 Effects of solo and joint action control on interpersonal neural correlations (n = 20 pairs).** The analyses used sliding window zero-lag Pearson correlations between the EEG time series at each electrode for the two participants within each pair. **a** Trial time-line. Correlations are time-referenced to the spatial action onset cue. The timing of trial events is depicted with filled circular markers. RT (action initiation) and RT+MT (action completion) markers reflect the mean of solo and joint trials. The variable delay prior to action onset (500–1000 ms) resulted in a range of onset timings for the fixation and action control cue stimuli; the corresponding markers reflect the mean onset timing across trials. The pink shaded area depicts time points where interpersonal neural correlations were statistically larger for joint compared with solo control. **b** Correlation heat map for solo control. For visualisation, the electrodes on the y axis have been sorted firstly anterior (ant.) to central (cent.) to posterior (post.) and secondly left to right. **c** Correlation heat map for joint control actions. **d** Thresholded correlations for the contrast joint minus solo. Grey bands indicate statistically higher interpersonal neural correlations for joint compared with solo control. The crosshair lines (red) in red indicate the peak difference at electrode Cz, which arose 1542 ms after the spatial action cue. **e** Correlation time series at electrode Cz. The pink shaded region indicates time points with significantly higher interpersonal correlations for joint compared with solo control. **f** Grand mean correlation subtraction topography at peak correlation (1542 ms). **g** Correlations at electrode Cz at peak correlation as a function of control condition (solo, joint) and trial accuracy (correct, error). **h** ERPs at electrode Cz time-locked to action offset. **i** Morlet wavelet of the grand mean action offset ERP.

To assess the statistical significance of the third band of interpersonally correlated neural activity, we submitted the correlational data to permutation simulations in which each trial was randomly assigned a control label (solo or joint) and the grand mean subtraction contrast (joint minus solo) was recomputed. Note that permutation simulations have the virtue of inherently controlling for multiple comparisons (here, across electrodes and time points), as each datapoint contributes to the null hypothesis distribution. With α set at 1e-04 (two-tailed), interpersonal neural correlations were significantly higher for joint compared with solo control, peaking late in the trial following action completion (Fig. 8d). A time series representation is provided in Fig. 8e, showing that interpersonal neural correlations peaked uniquely for joint control late in the trial (black line), and that the earlier peak preceding action initiation (i.e., RT; red and blue lines) did not differ significantly between solo and joint actions. A topographic time slice at peak correlation is illustrated in Fig. 8f, showing a central focus at electrode Cz, which extended across the left/right axis (electrodes C1/C2). Given that correlations at occipitoparietal (i.e., posterior) electrodes approach zero at the time point of peak correlation (Fig. 8b, c), these findings taken together suggest that the stronger correlation for joint control likely reflects a motoric rather than visual process[54,55].

*Joint but not solo behavioural performance predicts the magnitude of neural correlations.* To test whether increased interpersonal neural correlations during joint control were related to behavioural performance and thus visuomotor control processes, we examined correlation magnitude at the peak spatiotemporal slice (electrode Cz, 1542 ms post spatial cue onset) as a function of task performance (Fig. 8g). Interpersonal correlations (r) were submitted to a two-way repeated-measures ANOVA with the factors of control (solo, joint) and accuracy (correct, error). All effects were significant (control: $F_{1,19} = 42.70$, $p = 2.94e-06$, $\eta_p^2 = 0.692$; accuracy: $F_{1,19} = 15.34$, $p = 0.001$, $\eta_p^2 = 0.447$; control × accuracy: $F_{1,19} =$

coloured bands) were maximal at two time points, first during an early period coincident with the onset of the fixation placeholder and action control cue, and second during a later period following the onset of the spatial cue. The earlier period revealed maximal correlations at posterior (i.e., occipitoparietal) electrodes, consistent with visual ERPs to the onset of fixation and control cues. Interpersonal neural correlations at the later time period, following the spatial cue, were much more topographically dispersed, encompassing posterior, central, and anterior electrodes. This likely reflects coincident timing across co-actors of a number of temporally overlapping cognitive processes, including visual perception and action execution. Of primary interest was a third band of correlated activity, apparent only for joint control, that appeared later in the trial, peaking following action completion (i.e., experiment mean RT+MT; Fig. 8c).

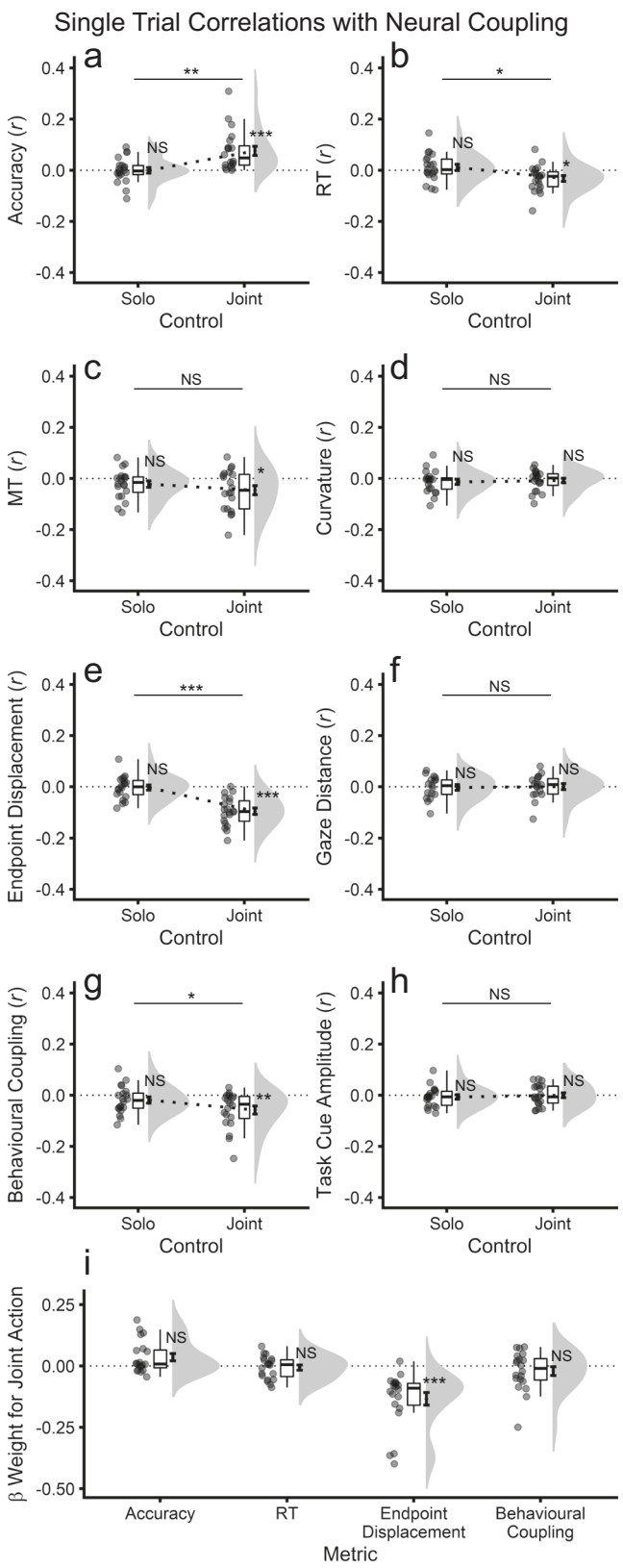

Single Trial Correlations with Neural Coupling

**Fig. 9 Single-trial level correlations between key task performance metrics and interpersonal neural coupling ($n = 20$ pairs).** All analyses were performed for visible rather than invisible cursors. Neural coupling was calculated at electrode Cz at 1542 ms post action-cue onset, which was the critical spatiotemporal slice for neural coupling determined in the main experiment-level analyses. For solo trials, accuracy, RT, MT, curvature and endpoint displacement were calculated as the mean of visible solo cursors. **a** Accuracy. **b** RT. **c** MT. **d** Curvature, calculated as the mean across trial frames. **e** Endpoint displacement. **f** Inter-gaze distance, calculated on the frame corresponding to the experiment mean action completion time (RT +MT: $478 + 662 = 1140$ ms). **g** Behavioural coupling, calculated as the absolute magnitude of the differences in action offset times between participants within each pair. **h** Task cue amplitude, calculated as the average Morlet wavelet amplitude across electrodes O1, Oz, O2 at the time point of peak difference between solo and joint control (1148 ms post task cue onset). **i** Pair-level coefficients of single-trial level multiple regression analyses for joint control trials.

between correct ($M = 0.01$, $SD = 0.03$) and error trials ($M = 0.01$, $SD = 0.03$; $t_{19} = 0.23$, $p = 0.882$). Additionally, single-sample $t$-tests showed that joint control interpersonal correlations were significantly greater than zero for both correct ($t_{19} = 6.13$, $p = 6.83$e-06) and error trials ($t_{19} = 4.11$, $p = 5.96$e-04). By contrast, solo interpersonal correlations did not differ significantly from zero (correct: $t_{19} = 0.99$, $p = 0.334$; error: $t_{19} = 1.14$, $p = 0.269$). These patterns link interpersonal neural correlations (i.e., neural coupling) specifically to successful joint control performance, closely mirroring the patterns of interpersonal behavioural correlations (cf. Figs. 8g and 5d). Together with the central topographic focus (Fig. 8f), the results suggest that neural coupling reflects task-related motoric coupling between the co-actors.

*Joint neural correlations reflect temporal alignment of action offsets between co-actors.* It might initially seem surprising that interpersonal neural correlations during joint control peaked after mean action completion time (1542 ms vs. 1115 ms; Fig. 8e). One might instead predict that such correlations should be more closely related to online action execution. Note, though, that there were in fact topographically widespread correlations during action execution (Fig. 8b, c), but these did not differ significantly between solo and joint control (Fig. 8d, e). Thus, neural correlations during action execution were attributable to simultaneous performance, rather than real-time coordination between co-actors. One possibility is that action completion produced an ERP component that was aligned for joint but not solo control. Consistent with this hypothesis, action offsets between co-actors were more highly correlated during joint compared with solo control (Fig. 5d). To assess this possibility further, we created trial-averaged ERPs time-locked to action offset for both solo and joint control. As illustrated in Fig. 8h, this analysis produced a measurable ERP at electrode Cz. The ERP waveforms were highly similar for joint and solo control. Importantly, for solo control, the ERP analysis used separate time-locking for each co-actor, whereas for joint control we used a common offset for both participants. This result provides a plausible hypothesis for interpersonal neural correlations: temporally coincident action offsets produced coincident ERPs for joint but not solo control since solo control offsets were not as closely aligned at the single-trial level. Additionally, time-frequency analyses on the grand mean action offset ERP showed maximum energy in the delta frequency range (<4 Hz; Fig. 8i), ruling out spatial attention to the target frequency (17/19 Hz) as a possible mechanism, which did not differ significantly between solo and joint control (Fig. 7).

13.49, $p = 0.002$, $\eta_p^2 = 0.002$). Follow-up within-subjects $t$-tests indicated that joint control interpersonal correlations were significantly higher on correct ($M = = 0.09$, $SD = 0.07$) compared with error trials ($M = 0.03$, $SD = 0.04$; $t_{19} = -4.15$, $p = 5.46$e-04), but that solo interpersonal correlations did not differ significantly

*Interpersonal neural coupling is uniquely correlated with endpoint displacement at the single-trial level.* The results presented above suggest that neural coupling reflects improved joint action performance. To assess this further, at the single-trial level, we correlated neural coupling with each of the performance metrics that differed significantly between joint and solo control, namely, accuracy, RT, MT, curvature, endpoint displacement, gaze distance, behavioural coupling, and task cue amplitude. We divided trials by action control (joint, solo) and averaged the resulting Pearson's correlation coefficients ($r$) across participant pairs. We performed three statistical tests: one repeated-measures $t$-test that contrasted mean joint and solo correlations, and two single-sample $t$-tests against the null hypothesis of $\mu = 0$ separately for solo and joint trials. As depicted in Fig. 9, neural coupling was selectively associated with joint control rather than solo control. Accuracy (Fig. 9a), RT (Fig. 9b), endpoint displacement (Fig. 9e) and behavioural coupling (Fig. 9g) each showed single-trial correlations with neural coupling that were significantly ($\alpha = 0.05$) different from zero for joint control ($|rs| \geq 0.03$, $t_{19}s > 2.83$, $ps \leq 0.011$) and significantly greater in absolute magnitude for joint compared with solo control ($|\Delta rs| \geq 0.02$, $t_{19}s \geq 2.60$, $ps \leq 0.012$). For solo control, no metrics were significantly correlated with neural coupling ($|rs| \leq 0.02$, $ts \leq 1.74$, $ps \geq 0.098$). MT showed a correlation for joint control that was significantly greater than zero ($r = 0.05$, $t = 2.59$, $p = 0.018$), but this correlation did not differ significantly in magnitude from the correlation for solo control ($\Delta r = 0.03$, $t = 1.06$, $p = 0.303$; Fig. 9c).

To further contrast accuracy, RT, endpoint displacement and behavioural coupling, these metrics were entered as predictors of neural coupling in standard multiple regression analyses. These analyses were performed for joint control at the single-trial level for each pair, with standardised coefficients ($\beta$ weights) averaged across pairs ($N = 20$). Using the conservative Bonferroni family-wise error rate correction ($\alpha = 0.0125$ for four metrics), only endpoint displacement showed a significant negative partial correlation with neural coupling ($\beta$: $M = -0.13$, $SD = 0.11$; $t = -5.27$, $p = 4.39e{-}05$), with smaller endpoint displacements reflecting greater neural coupling, uniquely explaining 1.79% of single-trial variance. This effect remained highly significant when excluding three outlying cases with the strongest coefficients ($\beta$: $M = -0.09$, $SD = = 0.05$; $t = -7.44$, $p = 1.41e{-}06$). RT, accuracy, and behavioural coupling did not significantly explain unique single-trial variance in neural coupling ($|\beta s|$: $Ms \leq 0.04$, $SDs \leq 0.11$; $ts \leq 2.42$, $ps \geq 0.026$; Fig. 9i). These single-trial level results further support the association between neural coupling and behavioural performance under joint control.

*Motoric similarity, not visual similarity, underpins interpersonal neural coupling.* The co-actors viewed identical displays during joint control but distinct displays during solo control. To assess whether increased neural coupling for joint control reflected visual similarity across the participants' displays, we calculated the maximum inter-cursor distance (ICD) between co-actors (see Supplementary Movie 3). On solo trials, ICD reflected the degree of visual similarity experienced by the co-actors, with smaller distances indicating more visually similar trials. By contrast, joint trials always had identical visual input due to the common cursor movement. The individual contributions underlying the shared visual input on joint trials were invisible solo cursors, and ICD here reflected motoric similarity between pair members (Fig. 10a).

ICD was submitted to a repeated-measures ANOVA with the factors of control (solo, joint) and accuracy (correct, incorrect). All effects were significant (control, accuracy, control × accuracy: $Fs_{1,19} > 53.75$, $ps < 5.98e{-}07$, $\eta_p^2 s > 0.739$). Follow-up tests showed that ICD (°) was significantly smaller (and thus visual similarity was significantly greater) for solo control when both participants

were individually correct ($M = 3.92$, $SD = 0.87$) versus incorrect (i.e., error trials in which one or both participants were incorrect; $M = 5.22$, $SD = 1.46$; $t_{19} = -5.42$, $p = 3.16e{-}05$; Fig. 10b). This was expected, as correct trials constrained the range of viable actions. Despite increased visual similarity on solo correct versus error trials, neural coupling did not differ significantly (Fig. 8g). Therefore, visual similarity cannot explain interpersonal neural correlations on solo trials. For joint control, as expected, the maximum ICD was significantly smaller for correct ($M = 6.53$, $SD = 1.92$) compared with error trials (with accuracy determined by the visible joint cursor's trajectory, ($M = 12.84$, $SD = 4.82$; $t_{19} = -8.30$, $p = 9.67e{-}08$; Fig. 10b). Thus, motoric similarity was greater for correct compared with error joint trials, and greater motoric similarity was associated with interpersonal neural correlations (Fig. 8g). Thus, motoric similarity between co-actors, rather than visual similarity between stimulus displays, explained the correspondence between neural coupling and successful joint action.

## Discussion

We sought to identify the cognitive stages and associated neural activity patterns underlying the performance benefits of redundant joint object control, which involves co-actors sharing the same action possibilities[17,22–27]. We measured behavioural performance, eye gaze, and neural activity as pairs of participants used video game controllers to guide a cursor to a target, while ignoring a concurrent distractor in the opposite visual field. Participants performed the task either individually (solo control) or together with a co-actor (joint control). Extending previous work, this study's comprehensive multi-modal spatiotemporal analyses at the neural and behavioural levels, at the individual and pair levels, and at the experiment and single-trial levels, provides a concrete mechanism for interpersonal neural coupling that excludes confounding factors of signal averaging and visual similarity, and links neural coupling specifically to real-time coordination between co-actors and successful and precise joint actions.

Our study identifies a direct brain-behaviour link underpinning neural coupling arising from coincident motor offsets between co-actors during joint control. First, behavioural coupling, defined as correlations in video game controller displacement across co-actors, was maximal at action completion and predictive of successful performance for joint but not solo control. Second, neural coupling, defined as correlations in EEG time series across co-actors, was again maximal at action completion and predictive of successful performance for joint but not solo control. Third, neural coupling was maximal at central electrodes implicating motoric rather than visual processes. Fourth, ERP analyses time-locked to action offset times revealed an action offset response that was temporally aligned under joint but not solo control. Fifth, behavioural and neural coupling were significantly correlated at the single-trial level for joint but not solo control. Sixth, neural coupling under joint action was uniquely correlated with endpoint displacement at the single-trial level. Finally, ICD of invisible cursors on joint trials was smaller and thus motor similarity was larger on correct compared with error trials under joint action, linking motoric similarity between co-actors with successful joint action. These converging results reveal a direct brain-behaviour link underpinning successful joint action performance.

Our study addressed a key empirical challenge to establish that joint action is attributable to real-time coordination between co-actors rather than simultaneous but independent task performance (i.e., signal averaging)[48,49]. In this regard, investigations of redundant control have the notable strength that task demands are matched between co-actors, who not only share action

## Visual Similarity Versus Motoric Similarity

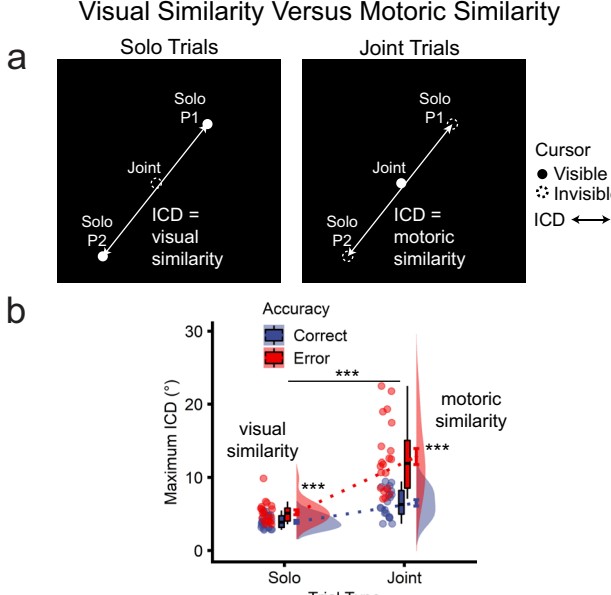

**Fig. 10 Comparison of motoric similarity and visual similarity as explanatory factors for the correspondence between behavioural performance and interpersonal neural coupling (n = 20 pairs). a** Depiction of maximum inter-cursor distance (ICD) between visible solo cursors on solo trials (reflecting visual similarity between co-actors) and invisible solo cursors on joint trials (reflecting motor similarity). **b** Effects of control and accuracy on ICD.

possibilities with each other, but also share possibilities across solo and joint trials. This feature of redundant control allows motoric demands and visual inputs to match closely for solo and joint control. Furthermore, we used a comprehensive analytic approach of contrasting the trajectories of visible and invisible cursors across a complete suite of behavioural performance metrics. These analyses showed that, first, performance was severely degraded for invisible compared with visible joint cursors and, second, that cursor control and cursor visibility factors interacted significantly for all key behavioural metrics. These results provide strong evidence that joint control depends on real-time coordination and not signal averaging. Real-time coordination was apparent as statistically greater neural coupling at action completion on joint compared with solo trials. By contrast, interpersonal neural correlations attributable to simultaneous performance occurred after the onset of the spatial cue, peaking at action onset time. Importantly, neural coupling was restricted to central topographic regions, implicating motoric processes[54,55]. By contrast, correlations attributable to simultaneous performance encompassed the entire topography along the posterior-anterior axis, implicating visual, motoric and potentially higher-order cognitive processes[39–41,56,57]. In summary, our use of a redundant control task, analyses of cursor visibility, and parallel behavioural and neural measurements allowed us to attribute joint action performance and neural coupling to real-time coordination between co-actors.

We addressed another empirical challenge to establish that neural coupling is attributable to motoric rather than visual similarity. First, at the single-trial level, neural coupling was correlated with endpoint displacement but not with gaze distance. Second, participants were not simply tracking the cursor, but instead made saccades toward the target position before initiating cursor movement. This suggests that the reduced inter-gaze distance on joint trials was not driven by a one-to-one

correspondence between the cursor and gaze positions. Third, joint control involved identical visual displays, and thus visual input cannot explain increased neural coupling for correct compared with error joint trials. Fourth, ICD of invisible solo cursors on joint trials reflected motoric similarity, which was greater for correct compared with error trials, consistent with increased neural coupling, which was larger on correct trials.

Underpinning neural coupling, our study identified a phase-locked electrophysiological component, which we refer to as an action offset ERP. This provides a direct mechanism for neural coupling: coordinated action offsets under joint control produced similarly timed action offset ERPs across co-actors, and thus, interpersonally correlated neural activity. Action offset times on solo trials were not correlated between co-actors, and therefore neither was the corresponding neural activity. Although previous vision and audition research has focused on onset rather than offset ERPs[58–60], pure offset ERPs have been documented[61–63], and frequency-tagged ERPs (e.g., SSVEPs) naturally depend on a combination of both onsets and offsets[47]. Motoric investigations have focused on neural activity preceding actions or corresponding to discrete keypresses[54,55,64]. In both cases, motor onsets and offsets merge into a single component. By contrast, our participants used video game controllers to initiate movement, guide the cursor to the target location and offset at a variable time, on average ~650 ms later, effectively decoupling action offsets and onsets. The results suggest that a defining feature of the action offset ERP is a slow temporal evolution with peaks at ~200, ~500, and ~800 ms, corresponding to a delta oscillation (<4 Hz). By contrast, separate onset/offset ERPs in the visual and auditory modalities and combined onset/offset ERPs in the motor domain are typically complete within ~500 ms, thereby producing theta oscillations (4–8 Hz)[65,66]. Consistent with previously documented motoric ERPs[54,55], the action offset ERP had a central topographic focus, which combined with the slow temporal evolution suggests a motoric rather than visual source. Most importantly, the action offset ERPs limit the correlation of interpersonal neural coupling to the delta frequency band.

Previous human and monkey investigations of redundant object control have suggested that groups may outperform individuals[17,22–27]. The current results show that joint control improved performance relative to LP solo control. Performance under joint control was equivalent to or worse than HP solo control, supporting race model accounts[48,49] of joint action, with performance accounted for by statistical facilitation – picking the higher fidelity motoric signals provided by the HP co-actor. The current results are consistent with previous reports that the LP co-actor benefits most from joint control[17,22,24], although those reports also showed a benefit, albeit a smaller one, for the HP solo co-actor. Other studies have shown benefits or equivalence for the group compared with the individual[23,25–27], although these studies generally categorised individuals (LP/HP) at the experiment level rather than the single-trial level. For the current trajectory analyses, we adopted the single-trial approach, which is a more stringent test of joint performance. Previous investigations of redundant control have focused on behaviour, although intriguingly, recent evidence from monkey neurophysiology has suggested the presence of "joint action" selective neurons in the dorsal premotor cortex[26]. This is consistent with the current study's result of neural coupling arising from motor cortical activity, and raises the possibility that the activity of joint action selective cells might be measurable, albeit at the population level and with reduced fidelity, within scalp-level EEG recordings.

The joint action paradigm of the current study was modelled on the approach of Visco-Comandini and colleagues[25], who used a similar paradigm to assess whether non-human primates

cooperate. The authors trained macaque monkeys to move a cursor to a target either individually or jointly. Their results showed that monkey pairs adjusted their individual performance during joint action, that joint cursor trajectories were less curved and slower compared with solo trajectories, and that differences in individual RTs between co-actors were smaller on joint compared with solo control trials. Their results are consistent with real-time coordination and with the current study. We extended the work of Visco-Comandini and colleagues[25], which focused on action initiation and execution, to encompass action preparation and completion periods. We also extended their analyses of behaviour to analyses of brain signals and brain-behaviour relationships. A follow-up study by Satta and colleagues[21] used a version of the same paradigm to assess the developmental trajectory of cooperation in children aged 6–9 years. The results showed improvements in joint action performance over this developmental period, including changes in real-time coordination, which the authors suggested reflect the emergence of the capacity for cooperation. It would be informative to know whether neural coupling between co-actors at action completion also develops during this age period.

It is noteworthy that the instantiation of interpersonal coordination as redundant joint object control is just one of many possible modes of interpersonal interaction. For instance, joint action has been variously defined as "the ability to interact with a partner in order to reach a common goal"[25], "the ability to coordinate our actions with those of others"[15,33], "how people manage to predict each other's actions"[16], "the development of a leader–follower relationship"[23], "action corepresentation"[67], and "physical assistance enabled by haptic interaction"[17], to list just a few definitions. During the review process it was asked whether the patterns of joint object control we describe here can really be classed as "coordination", since the co-actors' goals were common and there was little opportunity for each individual to learn about their partner's intentions or actions. It was suggested that the term "coordination" should be restricted to instances in which interpersonal interaction requires formation of a model of the partners' minds and actions. In response, we suggest that this perspective raises tractable empirical and theoretical questions for future work. For instance: to what extent does neural coupling under redundant joint object control emerge simply from chance pairings versus requiring learning? Does having a mental model of a partner's intentions change the nature of interpersonal adaptations during redundant joint object control and perhaps boost performance? What are the common cognitive mechanisms that best categorise different modes of interpersonal interaction? What role does the computer play in mediating joint interactions in the experimental context? Can neurofeedback of brain activity during joint action help to improve coordination between co-actors?

Recent technological advances have made simultaneous EEG recordings in interacting subjects, termed "hyperscanning"[29,30,42,44,51–53,67,68], tractable for scientific investigation. Broadly speaking, there have been two key hyperscanning approaches. The first approach uses elementary visuomotor cursor control games. The work of Astolfi et al.[51] is one example that compared interpersonal neural coupling in pairs of participants working independently versus cooperating under joint control. A second hyperscanning approach uses more complex social interactions. For instance, Kourtis et al.[42] had participants perform jointly coordinated hand gestures that could result in distinct partner-dependent postural configurations, therefore explicitly modelling social processes. Other investigations have combined these two approaches. For instance, Sinha et al.[52] had subjects engage in joint action games, reminiscent of Pong (a "table tennis" video game classic), either remotely, involving no social presence, or in adjacent rooms, thereby

involving social presence. Each of these approaches has tended to identify neural coupling as a hallmark of joint interactions. Our study is unique among these in closely specifying a direct brain-behaviour relationship that underlies successful joint action performance.

Traditional investigations of perception, attention, and action have focused on the individual in isolation from their social context. We have extended previous work to introduce an experimental paradigm and a comprehensive analytical strategy that allows the simultaneous tracking of behaviour and neural signals across co-actors, and the demarcation of cognitive processing into discrete, separable stages from preparation to completion. Using this approach, future work could examine joint actions in pairs of individuals with disabilities involving visuomotor control or social cognition or could compare differences in behaviour and associated brain activity for co-actors that are already skilled in performing actions together compared with those who are not. More broadly, a challenge for future work will be to design interpersonal interactions that promote the benefits and minimise the costs of joint action.

## Methods

**Participants**. The University of Queensland Human Research Ethics Committee approved the study protocol for this quantitative behavioural/neuroimaging experiment, and all participants provided informed written consent. Twenty pairs of participants (total $N = 40$) volunteered for the study and were recruited from The University of Queensland Psychology Research Participation Scheme. Zero participants dropped out/declined participation. Self-selection and other recruitment biases are not likely to have impacted the results. Participants were 18–34 years old ($M = 22.33$ years, $SD = 2.98$; 20 males, two left-handed) and reported no personal or familial history of photosensitive epilepsy or seizures, and therefore could safely view the flickering visual displays. Four pairs were both males and four pairs were both females. Participants were paid for three hours at a rate of $10/h. Pay was not contingent on performance. Participants were pseudo-randomly paired based on their timeslot for participation with the requirement that pairs not be acquaintances. Participants were instructed to perform the task as quickly and accurately as possible. Data collection started on 7 June 2016 and ended on 5 August 2016. During testing, no one was present besides the two participants and one to three researchers (required for the dual-EEG setup phase). The researchers were not blind to the experimental conditions and study hypotheses during data collection. It is unlikely that this affected the key results, which were based on correlations between behavioural and neural measures.

**Solo and joint control visuomotor action tasks**. The study consisted of 15 blocks of 64 trials (960 in total), divided equally into solo and joint control trials (480 trials each). The presentation order of solo and joint control trials was randomised within each block. There were four pairs of target/distractor positions (see Fig. 1d), which were presented on separate trials and were counterbalanced with control (solo and joint), flicker frequency (17 & 19 Hz) and task-relevance (target and distractor) within each block. Blocks were separated by enforced 20-s rest periods. Participants completed 16 solo and 16 joint control practice trials.

Each trial began with a blank rest display (1000 ms), followed by a pre-control cue period (576 ms) during which a grayscale placeholder at fixation and the target and distractor appeared. Then, the placeholder at fixation changed colour (1424 ms), cueing participants with 100% validity to perform either a solo or joint control action. Following the control cue, there was a variable delay (500–1000 ms) before the presentation of the spatial cue (a small white arrow) at fixation. Participants had 2500 ms to initiate and complete the action, which involved bringing the filled dot at the centre of the cursor annulus into the target (black annulus). Participants had a window 200–800 ms after spatial cue onset to initiate the action, and had 1500 ms to complete the action, requiring that they hold the cursor within the target annulus for 200 ms. At the end of each trial, feedback was displayed (500 ms). Correct trials were indicated by response speed (RT+MT) in milliseconds. Total trial duration was 6500–7000 ms. Participants were allowed to freely move their eyes throughout the experiment, and gaze position was recorded remotely at a sampling frequency of 120 Hz, as described in detail below.

Participants used wired video game controllers (Xbox One; Microsoft) and used the right thumb stick to move the cursor. In real-time, the controller provided separate horizontal ($x$) and vertical ($y$) displacements between 0 and 1, which were scaled separately into vertical and horizontal cursor movements by a constant of 7.5 pixels on each frame (29.9°/s). Thus, the maximum cursor displacement of 1.0 corresponded to a maximum cursor velocity of 29.9°/s. The action threshold was 10% of maximum controller displacement; smaller displacements resulted in a cursor velocity of zero. Under joint control, individual horizontal and vertical

velocities were separately averaged across the co-actors if both co-actors exceeded the action threshold. Otherwise, the joint cursor velocity was set to zero.

**Frequency tagging**. Placeholders were created by phase-scrambling chequerboards (Fig. 1c). Placeholder flicker was counter-phased using an interpolation approach[69]. Placeholders at the target and distractor locations flickered at unique frequencies (17 and 19 Hz) throughout the trial following rest (5500–6000 ms total), which were counterbalanced across target positions. Target/distractor flicker phases were aligned to spatial cue onset. The central placeholder flickered at 7 Hz and was initially coloured grey before becoming the red or green control cue, with flicker phase aligned to stimulus onset.

**Stimulus presentation**. Stimuli were presented at 144 Hz on two ASUS VG248QE LCD monitors synchronised in Eyefinity display mode (combined resolution 3840 × 1080). Stimuli were presented on a black background (RGB: 0, 0, 0) at a viewing distance of 57 cm using the Cogent 2000 Toolbox (http://www.vislab.ucl.ac.uk/cogent_2000.php) running in MATLAB R2014a (32-bit; Mathworks, Natick, MA) under Windows 7 (64-bit) on a Dell Precision T1700 PC.

Red (max. RGB: 255, 0, 0), green (max. RGB: 0, 168, 0) and grey/pre-control cue (max. RGB: 140, 140, 140) placeholders were matched for luminance (~62 cd/m2) using a photometer. Target/distractor placeholders were white. Placeholders/cursors subtended 5.0 × 5.0°. Peripheral targets/distractors were presented at an eccentricity of 8.0°. The target annulus/area subtended 2.0 × 2.0°. The spatial cue subtended 0.5 × 0.5°. Instructional text was presented in white (RGB: 255, 255, 255) 40-point Lucida Console font.

**Other external tools**. Trajectory curvature was calculated using a publicly available method based on fitting a circle and calculating the inverse of its radius (https://au.mathworks.com/matlabcentral/fileexchange/69452-curvature-of-a-1d-curve-in-a-2d-or-3d-space). EEG topographies were created using the EEGLAB (https://sccn.ucsd.edu/eeglab/index.php) *topoplot* function. Wavelet analyses were conducted using the Brainstorm (https://neuroimage.usc.edu/brainstorm/Introduction) *morlet_transform* function. Error shading was plotted using a publicly available method (https://au.mathworks.com/matlabcentral/fileexchange/27485-boundedline-m).

**EEG recording and pre-processing**. EEG was sampled at 2000 Hz from 61 scalp channels using a 64-channel amplifier and Ag/AgCl electrodes positioned according to the 10-10 system within a WaveGuard cap (ANT Neuro, Germany). Data were acquired and exported using ANT Neuro software. The reference electrode was positioned at CPz, and the ground electrode positioned between FPz and Fz. Electrode impedance was kept within the range of 0–40 kΩ. Pre-processing was conducted using Brain Electrical Source Acquisition software (MEGIS Software GmbH, Germany). Given that participants were free to move their gaze during the experiment, eye movement artefacts were topographically interpolated with 150 µV horizontal and 250 µV vertical eye action thresholds (BESA v.6.3, MEGIS Software GmbH, Munich, Germany)[70]. Data were filtered using high-pass (0.5 Hz), low-pass (45 Hz) and notch (50 Hz, 2 Hz-width) filters. Trial epochs containing voltage fluctuations > 100 µV on any channel were marked as artefacts and were excluded from analyses. EEG measurements were synchronised between individuals within each pair using parallel port triggers. Total EEG recording duration was < 2 h, including rests. EEG analyses were performed using custom-written methods. EEG analyses were originally performed at 2000 Hz. In the interest of computational efficiency, EEG data were downsampled to 256 Hz for the SSVEP analyses and 64 Hz for correlational analyses. Statistical patterns are consistent at the original sampling rate and with downsampling. The downsampled results are reported.

**Eye tracking**. Gaze position measurements were synchronised between individuals within each participant pair using serial port triggers. Gaze position was sampled at 120 Hz using an iView Red-m infrared eye tracker (SensoMotoric Instruments, Germany). Head position was stabilised using chin rests. Calibration was conducted with a 5-point fixation scheme. Due to eye tracking difficulties associated with vision correction, three participants were excluded from the eye gaze analysis. Eye tracking data were pre-processed by excluding periods around blinks (±200 ms) and rapid increases in pupil area (>99.99th percentile)[71].

**Statistics and reproducibility**. Analyses were performed using custom-written methods for MATLAB R2015a (64-bit), R (version 4.0.2 [64-bit]), Python 3.8.5 (64-bit), and IBM SPSS Statistics Version 21. Raincloud plots[72] were produced using R's ggplot2 library. All other plots were produced using ggplot2, MATLAB, or Python's matplotlib.pyplot module. All statistical tests were based on repeated-measures and were two-tailed. Analyses corresponding to Figs. 2–5 and 8–10 were conducted at the pair level (N = 20 pairs). Analyses corresponding to Figs. 6 and 7 were conducted at the individual level (N = 40 individuals). Permutation simulations inherently controlled for multiple comparisons across time and electrodes as appropriate. A standard multiple regression controlled for multiple comparisons across single-trial level measures. The Bonferroni correction for multiple comparisons was applied separately for each metric in follow-up tests of ANOVA main effects and interactions.

Four participants from three participant pairs were excluded for missing or inaccurate eye tracking data. These participants were included in all other analyses. Note that uncorrected acuity was not a requisite for participation in the study, and correction can produce inaccuracies in eye tracking. Such exclusions are commonplace in eye tracking research[73], and do not affect the main conclusions of the study, which are based on EEG and behaviour.

The sample size (N = 20 pairs) and trial numbers were based on previous frequency-tagging investigations of attentional processes[45,65,68]. The analyses were verified in their entirety from the raw data to final summary statistics. Thus, the results were internally reproducible. All results correspond to statistical patterns pertaining to the full dataset.

**Reporting summary**. Further information on research design is available in the Nature Research Reporting Summary linked to this article.

## Data availability
This study's behavioural and processed eye tracking data are publicly accessible on The Open Science Framework (OSF; https://osf.io/yrbkx/?view_only=477894311aed41028deb49e29cd67d48 [accession code: yrbkx]). Due to OSF file and project size limits, the full dataset, including processed and raw eye tracking and EEG data and also final and intermediate results [282 GB] is available on the The University of Queensland Research Data Manager [accession code: JOINTACT17-A4673]) upon request. All source data underlying the graphs presented in the main figures is available in .csv format and can be plotted using the accompanying *supplementary_information.py* and *common.py* Python scripts (see Supplementary Data 1).

## Code availability
This study's game code and analysis code is publicly accessible on The Open Science Framework (https://osf.io/yrbkx/?view_only=477894311aed41028deb49e29cd67d48 [accession code: yrbkx]), GitHub (https://github.com/davidrosspainter/Joint-Action), and Zenodo (https://zenodo.org/record/4731244#.YIzliqERWLS)[74]. The code is also available on The University of Queensland Research Data Manager [accession code: JOINTACT17-A4673]).

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

## Acknowledgements

D.R.P. was supported by the Australian Research Council (ARC) Special Research Initiative Science of Learning Research Centre (SR120300015). J.B.M. was supported by

an ARC Australian Laureate Fellowship (FL110100103) and the ARC Centre of Excellence for Integrative Brain Function (ARC Centre Grant CE140100007)

## Author contributions

D.R.P., J.J.K., A.I.R. and J.B.M. conceived and designed the experiment. D.R.P. programmed the experiment. J.J.K. recruited participants. J.J.K., A.I.R. and D.R.P. collected the data. D.R.P., A.I.R. and J.J.K. analysed the data. D.R.P. and J.B.M. wrote the manuscript. J.J.K. and A.I.R. commented on the manuscript.

## Competing interests

The authors declare no competing interests.
