## [Peer Review File · Communications Biology]

Reviewers' comments:

Reviewer #1 (Remarks to the Author):

This manuscript aims to examine what underlies the performance benefits of motor coordination between two individuals at the levels of behavioral, cognitive, and neural processes. To this end, the authors performed a multi-modal experiment combining a computerized coordinated visuomotor task, eye tracking, and EEG. The manuscript presents results from analyses on behavioral and neural couplings during the joint action condition of the task, which the authors interpreted to be the signatures of the performance benefits of coordinated action.

This study attempts to follow up on a stream of recent literature on action coordination and its neural basis. The major issue I have is whether the experimental design really allows the authors to pursue the question they are interested in. Importantly, in the "joint" condition of the task, subjects receive visual feedback (i.e. display of the current cursor location) based on the mean of the inputs from the two players, without any information of the individual inputs. Therefore, this precludes any possibility of learning about what/how the other player is doing and adjust one's own action accordingly, which is what coordination is at its core. In other words, the "joint" condition of the task could be seen more or less as a solo task with distorted visual feedback. The empirical evidence about behavioral or attention differences between the joint and the solo condition in the manuscript is not sufficient to show that the players were indeed coordinating (partly due to another problem I will elaborate on in the next paragraph). A study on coordinated action must be able to show that one player is incorporating the other's action (or prediction thereof) into planning their own actions.

The setup of the "joint" condition also introduces a serious confound that is hard to control for. Unlike in the solo condition where there was some divergence, the pair of subjects viewed exactly the same visual stimuli in the joint condition, so there might be no surprise that their gaze patterns were more alike (Fig. 4) and their maneuvering of the controllers were more correlated (Fig. 5). It is possible that these could be primarily driven by the uniform visual feedback itself and therefore not related to any "coordination" per se. Similarly, while in the solo condition the two subjects likely acquired the target in different times (or even one might succeed while the other fails), in the joint condition the outcome was always the same for both, and target acquisition (and hence the participants letting go of the controller) also occurred at the same time. To me, this difference in external events/stimuli would be a more parsimonious explanation of the neural differences between conditions (Figs 6-8).

Notwithstanding my criticisms on the task design above, I am also not convinced that many of behavioral differences shown in Figs 2 and 3 are not due to simple signal averaging in the joint condition. Overall, any evaluation of the coordination (behavioral or neural) must use a hypothetical averaging of two solo trajectories as the baseline. Only by doing this can one isolate the pure effect of coordination above and beyond averaging. The only analysis in this manuscript that has this flavor is the one described in Lines 189-201 and Figs 3g and 3h. Even this analysis is not satisfying, as how accuracy is computed in this scenario is unclear. Conceivably, one player might have started and finished earlier than the other. What would be the average be in those time points when there was only one player? The authors should also perform more rigorous analyses on other metrics of performance using the same approach. For example, why not perform the same analysis as in Figs 3a to 3f for the hypothetical joint trajectory?

Finally, there is also more work to be done to more clearly specify what novel contributions this study brings. In particular, simultaneous EEG recordings in interacting subjects during a joint action task and neural correlations between them have been reported in some existing studies, such as:

Astolfi, L., Toppi, J., Vogel, P., Mattia, D., Babiloni, F., Ciaramidaro, A., & Siniatchkin, M. (2014, August). Investigating the neural basis of cooperative joint action. An EEG hyperscanning study. In 2014 36th Annual International Conference of the IEEE Engineering in Medicine and Biology Society (pp. 4896-4899). IEEE.

Sinha, N., Maszczyk, T., Wanxuan, Z., Tan, J., & Dauwels, J. (2016, October). EEG hyperscanning study of inter-brain synchrony during cooperative and competitive interaction. In 2016 IEEE

Some other intermediate and minor comments:

Quite a few of important details in the methods are not clearly spelled out (at least for me):

- a. Was there any monetary incentive for the subjects to make correct responses?
- b. How was action initiation and RT measured and defined for joint trials since the two players might initiate action in slightly different times?
- c. Why is action initiation under 200ms considered "too fast" and an error?
- d. What exactly are the controllers/joysticks controlling? What does controller displacement correspond to? I can imagine that the direction will correspond to the momentary direction of movement, but does the size of the displacement correspond to velocity or acceleration or anything else? Related to this question, what exactly are being averaged in the joint condition, location or the vector of movement?
- e. The instructions to the participants, especially what they were told about the joint condition, should be included.

Reviewer #2 (Remarks to the Author):

In this study the authors explore the neural processes underlying inter-individual motor coordination. To this purpose the EEG signals were recorded from subjects performing an isometric task, consisting in manoeuvring a cursor to a cue target location, under two different conditions, individually (SOLO control, SC) or together with a partner (JOINT control, JC). In the JC, the cursor was placed at the midpoint of the inputs received from the two co-actors. Twenty pairs of participants have been included in this experiment, and within each dyad the brain activity has been recorded simultaneously from the two interacting brains. At the behavioural level, the authors have found that the cursor trajectories were more accurate during the JC relative to SC, and this phenomenon was associated to interindividual coupling. Measuring the neural correlations between subject of each pair, the authors found a band of correlated activity emerging only during joint performance.

This is a well designed study, consisting in one of the few examples currently available in literature, in which the EEG activity is co-registered in subjects engaged in a dyadic joint-action context. The authors have adopted an appropriate experimental protocol to address the issue related to the understanding of the neural processes subtending interindividual action coordination. The statistical analysis is generally well performed and quite detailed, even if some more explanations are still needed.

I have few suggestions and comments that might be considered to improve the quality of the manuscript.

In the abstracts there are some sentences that need to be rephrased, such as

- The term "target acquisition" is not very clear (and this is true not only in the Abstract): since this term refers to a critical time, how is the "acquisition time" defined ?
- In the sentence "behavioural and neural coupling - quantified as increased interpersonal correlations...", please explain if "quantified as " refer to both behavioural and neural coupling
- Even the sentence "novel action offset neural response....interpersonal neural coupling" needs to be better explained.

Introduction:

p. 2 l. 26 when referring to "redundant object control", Bosga & Meulenbroek, 2007 should be cited.

It is quite clear that the review by Wahn, Karlinsky et al. 2018, has been for the authors a great source of inspiration, but it is too much emphasized. Original works should be cited, instead.

p. l. 61 and l. 63 When referring to "Previous studies", it should be stressed "Previous studies in humans", since studies in non-human primates have been performed indeed in a similar task to the one presented in the present manuscript, and addressing similar issues to those targeted by

the present manuscript.

Methods:

- One limit appears in how the behavioural stages are subdivided and consequently analysed. In particular, the authors restrict their attention on RT+MT, instead of RT and MT separately. I suggest to either explained the rationale of this choice, or even better to analyse them, as two distinct epochs. Furthermore, explain in more details, if errors are defined respect RT+MT or only to RT.
- How has the "action onset" been defined? Please explain.

Results:

One main concern I have about the results, regards the interindividual or better inter-dyadic differences observed in this study. Mainly grand averages have been reported, while the results would be stronger if evidence about the variability across individuals and pairs, both at behavioural and neural level, would be given.

- p. 7 l. 154 The sentence " with a tendency to resemble the HP rather than LP co-actor" does not seem to be supported by any particular evidence, so it should be omitted.
- Fig. 2 : I am not fully convinced about the results presented on Fig. 2a and Fig. 2c. According to what is shown on panel b and d, it seems that it is not correct to say that the accuracy in SC is smaller than JC, or that MT is longer in SC relative to JC. In fact, it depends on the subjects, as it is clearly shown in the respective panel b and d.
- Fig. 2: I do not understand why in Fig. 2f the authors do not use the same criterion to show the data for RTs, as for MT (Fig. 2d) and accuracy (Fig. 2b)
- Fig. 5. Data presented in this figures are difficult to interpret since not enough explanations are provided about how controller displacements have been measured.
- In Fig. 6c please report the time events in relation to the behaviour.

Finally, the Discussion is sometime redundant and it could be shortened. Instead of listing again the results, a greater effort should be made to discuss them in relation to the current literature in the field. The neural data, in particular, could be commented in relation to the other studies in which the brain activity has been recorded in a joint-action context. As an example, the present data could be related to the findings by Ferrari-Toniolo et al. 2019, in which the neural activity from frontal cortex in monkeys has been recorded under similar conditions as those adopted in the present study, or by Kourtis et al. 2019, in which EEG signals have been recorded in pairs of individuals performing a coordination task.

MINOR POINTS

A careful check is needed on the figures' order and numbering (for example Fig. 1d is cited before Fig. 1c, or Fig. 7c described before Fig. 7a)

Reviewer #3 (Remarks to the Author):

The authors present an interesting study wherein they have participants move an indicator to a cued target location, either in isolation or jointly with another participants (movement is the mean of both participants in the latter case). Concurrently with behavior, the authors measure eye-movements and EEG. Together, the results suggest that endpoints were more accurate under the joint than the solo condition, and this benefit is seemingly indexed by a coupling between participants. This coupling is shown in behavior by greater explained variance in joint than solo conditions (both LP and HP), increased similarity in gaze-position, greater offset correlation specifically in correct (vs. error trials), and in EEG via increased sensor-wise correlation in correct than incorrect trials during the joint condition.

The study is complete, leveraging a number of techniques, and clearly written. The statistical treatment also appears to be sound, and the illustrations provided appropriately convey the main findings. Overall, I am supportive of this manuscript.

I find that the manuscript could become stronger by performing correlational analyses, particularly

across trials and within subjects. The authors make a strong claim that behavioral and EEG coupling is leading to increase performance, and this is supported in large by effects that are specific to correct trials. But the authors could also show that trials in which coupling is greater, performance is best. Partial correlations where movement time, reaction time, curvature (etc.) are all controlled for one another while trying to account for accuracy could be interesting too. Likely movement duration and reaction time, for instance, would be correlated, but one can control for these correlations while trying to indicate which if these variables is important in increasing performance.

I am curious about the LP and HP categorization. Was this done as a whole, for all variables (for instance, RT and MT), or could Participant A be LP for MT and HP for RT? Similarly, was this done on the average of these variables, or on a trial-by-trial basis? The results presented in Figure 2 are not surprising, LP < joint < HP. Much more interesting, in Figure 3 we see that curvature is LP < HP < joint. If the authors could show that this is true even when allowing LP and HP to vary on a trial by trial basis, in effect the authors would have demonstrated that joint behavior can beat the race-model, statistical facilitation. At the moment it is unclear to me whether the sum is truly greater than its parts, as this has to be demonstrated by beating the race-model.

Endpoints become more accurate under the joint condition, and the authors suggest that trajectories become more linear. This is quantified by the appropriateness of a linear fit. I suggest the authors could more directly calculate curvature (angular velocity over radial velocity) to quantify curvature. This measure would also allow to examine whether the strategy of participants change in solo and joint conditions (for instance, up-fronting any needed curvature within a trajectory). Similarly, this analysis can be conducted on every trial, and thus the authors could truly examine the relation between curvature and endpoint accuracy. At the moment, the authors are only showing that both these measures change during the joint condition, but there is no sense of one of these variables driving the other.

Minor comments:

I would probably choose a different word than "manoeuvre" (used on a number of occasions), as many take this turn to refer to the act of moving around something. In this case there is no obstacle between the cursor and the target.

On a number of occasions the authors mention that social interaction is needed for cognition. I would suggest tempering this claim, as this is very much an active area of research.

In several figures the authors used a great number of asterisks (*) to denote significance. Personally to me, it is distracting to see a dozen of these asterisk and likely no reader will count how many there are. I would suggest the authors pick of cut-off, for example $p < 0.001$, and denote these with ***

The timing information in Figure 6 was not as clear to me as it was in the other figures. Please pick a more descriptive label for the x-axis in the ERP figure.

Response to Referees

We thank Reviewers 1, 2, and 3 for their detailed, thoughtful and positive appraisals of our submission. Below we outline our responses to each of the points raised in blue font.

Reviewer #1 (Remarks to the Author):

- (1) The major issue I have is whether the experimental design really allows the authors to pursue the question they are interested in. Importantly, in the “joint” condition of the task, subjects receive visual feedback (i.e. display of the current cursor location) based on the mean of the inputs from the two players, without any information of the individual inputs. Therefore, this precludes any possibility of learning about what/how the other player is doing and adjust one’s own action accordingly, which is what coordination is at its core. In other words, the “joint” condition of the task could be seen more or less as a solo task with distorted visual feedback. The empirical evidence about behavioral or attention differences between the joint and the solo condition in the manuscript is not sufficient to show that the players were indeed coordinating (partly due to another problem I will elaborate on in the next paragraph). A study on coordinated action must be able to show that one player is incorporating the other’s action (or prediction thereof) into planning their own actions.

We appreciate Reviewer 1’s point here, and draw attention to the fact that there is no single widely agreed definition of ‘joint action’. Reviewer 1’s description is one of many possible interpretations of the concept. For instance, joint action has been variously defined as “*the ability to interact with a partner in order to reach a common goal*”¹, “*the ability to coordinate our actions with those of others*”^{2,3}, “*how people manage to predict each other’s actions*”⁴, “*the development of a leader–follower relationship*”⁵, “*action corepresentation*”⁶ and “*physical assistance enabled by haptic interaction*”⁷, to list just a few definitions.

In this study, we sought to identify the cognitive and neural processes associated with redundant joint object control. In their recent review, Whan et al.⁸ define joint object control as occurring “*When two or more individuals **control an object together** [...] Redundant control refers to tasks where co-actors have the same action possibilities. For example, both co-actors can control object movement along the horizontal and vertical dimensions [...]. Note that in all of the studies that we consider in the present review, participants had*

visual access to the controlled object such that they could observe the combined effects of their own and their co-actor's actions on the object." Therefore, redundant joint control involves visual feedback of the combined effects of co-actions. Visual feedback of individual actions is not required, although some researchers have opted for this approach⁶.

Reviewer 1 suggests that *"without any information of the individual inputs [...] this precludes any possibility of learning about what/how the other player is doing and adjust one's own action accordingly, which is what coordination is at its core."* We politely disagree. During joint action, participants learn that the outcome is not determined solely by their own actions, and that their co-actor's actions contribute to a greater or lesser extent. Indeed, as we show, participants learn how to control their actions to an extent that they become, on average over the course of the experiment, faster and more accurate in completing actions compared with the low performing co-actor in each pair.

Visual feedback during joint action is accurate. Reviewer 1 suggests that *"[...] the 'joint' condition of the task could be seen more or less as a solo task with distorted visual feedback."* Under joint control, the visual feedback accurately reflects the combined motor actions of the two co-actors, and is not distorted. Instead, the individual's personal control over the cursor is diminished, consistent with the definition of Whan et al.⁸ (cited above) that joint action involves controlling an object together.

Reviewer 1 states: *"A study on coordinated action must be able to show that one player is incorporating the other's action (or prediction thereof) into planning their own actions."* As noted above, this depends on the definition of joint action one wishes to consider. Here we focused on redundant joint object control, and the results show that each member of a cooperating pair incorporates their partner's actions into their own, leading to faster and more accurate joint action control relative to LP solo control.

- (2) The setup of the "joint" condition also introduces a serious confound that is hard to control for. Unlike in the solo condition where there was some divergence, the pair of subjects viewed exactly the same visual stimuli in the joint condition, so there might be no surprise that their gaze patterns were more alike (Fig. 4) and their maneuvering of the controllers were more correlated (Fig. 5). It is possible that these could be primarily driven by the uniform visual feedback itself and therefore not related to any "coordination" per se. Similarly, while in the solo condition the two subjects likely acquired the target in different

times (or even one might succeed while the other fails), in the joint condition the outcome was always the same for both, and target acquisition (and hence the participants letting go of the controller) also occurred at the same time. To me, this difference in external events/stimuli would be a more parsimonious explanation of the neural differences between conditions (Figs 6-8).

We appreciate the opportunity to address this important point. Our findings, together with new analyses presented in the revised manuscript, demonstrate that neural coupling is explained by motoric rather than mere visual similarity on joint trials. We agree that, *prima facie*, eye gaze patterns (**Fig. 4**) may be explained in terms of a common visual input on joint action trials, since eye gaze could be a consequence rather than a cause of cursor movement. For controller correlations considered in isolation (**Fig. 5**), these might be attributable to shared visual input and/or real-time coordination between co-actors, since controller displacements are related both to joint action control and to the common visual input. Importantly, however, participants were not simply tracking the cursor, but instead made saccades toward the target position before initiating cursor movement (**Fig. 4e-f, Supplementary Video S2**). This suggests that the reduced inter-gaze distance on joint trials is not driven by a one-to-one correspondence between the cursor and gaze positions. (As a brief aside, the increased SSVEP amplitudes to the task cue on joint-action trials occurred prior to movement onset when stimuli were statically presented on both solo and joint trials, and thus cannot be attributed to differences in shared visual input between control conditions [**Fig. 6**]. Likewise, there were no effects of joint action control on spatial attention to the targets or distractors [**Fig. 7**]).

Two new analyses strongly suggest that one of our key results – neural coupling on joint trials – reflects motoric rather than visual similarity. First, neural coupling was more strongly associated with endpoint accuracy than gaze distance (**Fig. 9e** vs. **Fig. 9f**). Second, inter-cursor distance of solo trajectories was smaller for correct compared with error trials, and this effect was stronger on joint trials, mirroring the neural coupling results (**Fig. 10**). The rationale and results of these new analyses, including new figures, are outlined in the results sections titled “**Overall Task Performance: Overview**”, “**Neural coupling is uniquely correlated with endpoint accuracy at the single-trial level**”, and “**Motoric similarity, not visual similarity, underpins neural coupling**”.

- (3) Notwithstanding my criticisms on the task design above, I am also not convinced that many of behavioral differences shown in Figs 2 and 3 are not due to simple signal averaging in the joint condition. Overall, any evaluation of the coordination (behavioral or neural) must use a derived averaging of two solo trajectories as the baseline. Only by doing this can one isolate the pure effect of coordination above and beyond averaging. The only analysis in this manuscript that has this flavor is the one described in Lines 189-201 and Figs 3g and 3h. Even this analysis is not satisfying, as how accuracy is computed in this scenario is unclear. Conceivably, one player might have started and finished earlier than the other. What would be the average be in those time points when there was only one player? The authors should also perform more rigorous analyses on other metrics of performance using the same approach. For example, why not perform the same analysis as in Figs 3a to 3f for the derived joint trajectory?

Again, we thank Reviewer 1 for this suggestion. We agree it is imperative to ascertain whether the cursor trajectories and resulting behavioural patterns are attributable to real-time coordination between co-actors or signal averaging. Our design of solo versus joint control allows us to address this directly. We have redesigned the analyses of the behavioural metrics to demonstrate that real-time coordination, rather than signal averaging, underlies joint action. For this purpose, we have introduced a new factor, cursor visibility, which allows us to assess whether individual performance changes across joint and solo trials. Furthermore, we have improved our trajectory analyses to directly calculate trajectory curvature and endpoint accuracy. We have presented all results for the lower performing (LP) and higher performing (HP) pair members to more clearly contrast joint and solo control. These improved analyses are reported in detail in the sections **“Overall Task Performance: Overview”**, **“Joint control is attributable to real-time coordination rather than signal averaging”**, **“Compared with LP solo control, joint control improved accuracy and MT but slowed RT”**, **“Joint control reduced trajectory curvature and increased endpoint precision for joint control compared with LP solo control”**, and **“Motoric similarity, not visual similarity, underpin neural coupling”**.

- (4) Finally, there is also more work to be done to more clearly specify what novel contributions this study brings. In particular, simultaneous EEG recordings in interacting subjects during a joint action task and neural correlations between them have been reported in some existing

studies, such as:

Astolfi, L., Toppi, J., Vogel, P., Mattia, D., Babiloni, F., Ciaramidaro, A., & Siniatchkin, M. (2014, August). Investigating the neural basis of cooperative joint action. An EEG hyperscanning study. In 2014 36th Annual International Conference of the IEEE Engineering in Medicine and Biology Society (pp. 4896-4899). IEEE. Sinha, N., Maszczyk, T., Wanxuan, Z., Tan, J., & Dauwels, J. (2016, October). EEG hyperscanning study of inter-brain synchrony during cooperative and competitive interaction. In 2016 IEEE International Conference on Systems, Man, and Cybernetics (SMC) (pp. 004813-004818). IEEE.

We have revised the **Discussion** to better relate the results to the current literature in the field. As requested, we have cited and discussed the studies of Astolfi et al.⁹ and Sinha et al.¹⁰

What exactly are the controllers/joysticks controlling? What does controller displacement correspond to? I can imagine that the direction will correspond to the momentary direction of movement, but does the size of the displacement correspond to velocity or acceleration or anything else? Related to this question, what exactly are being averaged in the joint condition, location or the vector of movement?

The displacement directly mapped to cursor velocity. We have clarified in the revised manuscript as follows (**Methods: Solo and joint control visuomotor action tasks**): “In real-time, the controller provided separate horizontal (x) and vertical (y) displacements between 0 and 1, which were scaled separately into vertical and horizontal cursor movements by a constant of 7.5 pixels on each frame (29.9°/s). Thus, the maximum cursor displacement of 1.0 corresponded to a maximum cursor velocity of 29.9°/s. The action threshold was 10% of maximum controller displacement; smaller displacements resulted in a cursor velocity of zero. Under joint control, individual horizontal and vertical velocities were separately averaged across the co-actors if both co-actors exceeded the action threshold. Otherwise, the joint cursor velocity was set to zero.”

- (5) The instructions to the participants, especially what they were told about the joint condition, should be included.

Are two heads better than one?

We have now included more detailed instructions to participants (**Results: Task Overview**): “The action control contingency was made explicit to participants [...] Participants were instructed to emphasise both the speed and precision of their solo and joint control actions, and were not permitted to communicate with each other. Participants were told they would share cursor control on joint trials, and that to be successful, they would have to work cooperatively with their partner.”

Reviewer #2 (Remarks to the Author):

This is a well-designed study, consisting in one of the few examples currently available in literature, in which the EEG activity is co-registered in subjects engaged in a dyadic joint-action context. The authors have adopted an appropriate experimental protocol to address the issue related to the understanding of the neural processes subtending interindividual action coordination. The statistical analysis is generally well performed and quite detailed, even if some more explanations are still needed.

We thank Reviewer 2 for this positive appraisal of our work.

- (1) In the abstract there are some sentences that need to be rephrased, such as
 - The term “target acquisition” is not very clear (and this is true not only in the Abstract): since this term refers to a critical time, how is the “acquisition time” defined?

To improve clarity, throughout the manuscript we have replaced the term “target acquisition” with “action completion”, which we hope is more self-explanatory. Additionally, we have now provided the following definition (**Results: Task Overview**): “Actions were considered complete/accurate when the cursor first entered and remained within the target”. Note also that the following description is provided (**Joint control increases neural correlations between co-actors**): “Of primary interest was a third band of correlated activity, apparent only for joint control, that appeared later in the trial, peaking following action completion (i.e., mean RT + MT; **Fig. 8c**).

- (2) In the sentence “behavioural and neural coupling - quantified as increased interpersonal correlations...”, please explain if “quantified as” refer to both behavioural and neural coupling

We have clarified as follows (**Abstract**): “Joint control involved increases in *both* behavioural and neural coupling – *both* quantified as interpersonal correlations – peaking at action completion.” (emphasis added)

- (3) Even the sentence “novel action offset neural response...interpersonal neural coupling” needs to be better explained.

We have clarified as follows (**Abstract**): “Correspondingly, a neural offset response acted as a mechanism for and marker of interpersonal neural coupling, underpinning successful joint actions.”

- (4) p. 2 l. 26 when referring to “redundant object control”, Bosga & Meulenbroek, 2007 should be cited.

It is quite clear that the review by Wahn, Karlinsky et al. 2018, has been for the authors a great source of inspiration, but it is too much emphasized. Original works should be cited, instead.

We have now cited Bosga and Meulenbroek¹¹. We have also added and discussed additional original works including Astolfi et al.⁹, Sinha et al.¹⁰, Ferrari-Toniolo et al.¹², and Kourtis et al.¹³.

- (5) p. l. 61 and l. 63 When referring to “Previous studies”, it should be stressed “Previous studies in humans”, since studies in non-human primates have been performed indeed in a similar task to the one presented in the present manuscript, and addressing similar issues to those targeted by the present manuscript.

As requested, we have now specified (**Introduction, Discussion**) “previous human studies”, “previous human investigations” and “previous human findings”.

- (6) One limit appears in how the behavioural stages are subdivided and consequently analysed. In particular, the authors restrict their attention on RT+MT, instead of RT and MT separately. I suggest to either explained the rationale of this choice, or even better to analyse them, as two distinct epochs. Furthermore, explain in more details, if errors are defined respect RT+MT or only to RT.

In fact, RT and MT were and are separately analysed under the sections titled “**Joint control is attributable to real-time coordination rather than signal averaging**” and “**Compared with LP solo control, joint control improved accuracy and MT but slowed RT**”. These results are

plotted in **Fig. 2**. Additionally, RT and MT are separately analysed in a new section titled “**Neural coupling is uniquely correlated with endpoint displacement at the single-trial level**”. These new results are plotted in **Fig. 9**. Note that it is appropriate to describe action offsets as occurring at the time given by the sum of RT and MT (i.e., RT+MT) as depicted in **Fig. 8a**. Interpersonal neural correlations did not differ significantly between solo and joint control at the timepoint corresponding to mean RT (**Fig. 8a-e**).

Errors were calculated with respect to RT and MT separately. A trial could be incorrect for violating the RT criterion, the MT criterion, or both. We have clarified the classification of error trials in the section titled “**Overall Task Performance: Overview**” (paragraph 3).

- (7) How has the "action onset" been defined? Please explain.

We have clarified in the **Fig. 5** legend as follows: “Onsets for each trial were calculated separately for each co-actor on solo and joint trials as the first frame of movement showing displacements greater than 10% of maximum.”

- (8) One main concern I have about the results, regards the interindividual or better inter-dyadic differences observed in this study. Mainly grand averages have been reported, while the results would be stronger if evidence about the variability across individuals and pairs, both at behavioural and neural level, would be given.

We agree and have now provided information about variability throughout the revised manuscript using raincloud plots, at the behavioural and neural levels, at the individual and dyadic levels, and at the experiment and single-trial levels. We have replaced all the bar plots in **Figs. 2, 3, 4, 5, 6, 7, and 8**, and in new **Figs. 9 and 10**.

- (9) p. 7 l. 154 The sentence “with a tendency to resemble the HP rather than LP co-actor” does not seem to be supported by any particular evidence, so it should be omitted.

We agree and have removed this phrase.

- (10) Fig. 2: I am not fully convinced about the results presented on Fig. 2a and Fig. 2c. According to what is shown on panel b and d, it seems that it is not correct to say that the accuracy in SC is smaller than JC, or that MT is longer in SC relative to JC. In fact, it depends on the subjects, as it is clearly shown in the respective panel b and d.

We agree and have removed analyses comparing mean solo and joint performance in favour of presenting only individualised solo performance (i.e., HP vs. LP solo).

- (11) Fig. 2: I do not understand why in Fig. 2f the authors do not use the same criterion to show the data for RTs, as for MT (Fig. 2d) and accuracy (Fig. 2b)

We agree and have now calculated RT in a manner that is consistent with that of accuracy and MT.

- (12) Fig. 5. Data presented in this figure are difficult to interpret since not enough explanations are provided about how controller displacements have been measured.

As requested, we have provided more on the calculation of these metrics in the **Fig. 5** legend.

- (13) In Fig. 6c please report the time events in relation to the behaviour.

With arrows and labels, we have updated the timeline in **Fig. 6a** and **Fig. 6c** to mark the timeline as beginning at “control cue onset” and ending at “control cue offset”. Additionally, we have updated the corresponding **Fig. 6** legend to include the following statement: “Depicted data refer to the trial period beginning with action cue onset and ending with action cue offset.”

- (14) Finally, the Discussion is sometime redundant and it could be shortened. Instead of listing again the results, a greater effort should be made to discuss them in relation to the current literature in the field. The neural data, in particular, could be commented in relation to the other studies in which the brain activity has been recorded in a joint-action context. As an example, the present data could be related to the findings by Ferrari-Toniolo et al. 2019, in which the neural activity from frontal cortex in monkeys has been recorded under

Are two heads better than one?

similar conditions as those adopted in the present study, or by Kourtis et al. 2019, in which EEG signals have been recorded in pairs of individuals performing a coordination task.

We have revised the **Discussion** to better relate the results to the current literature in the field. As requested, we have cited and discussed the studies of Ferrari-Toniolo et al.¹² and Kourtis et al.¹³.

MINOR POINTS

- (15) A careful check is needed on the figures' order and numbering (for example Fig. 1d is cited before Fig. 1c, or Fig. 7c described before Fig. 7a)

We have updated the text and figures to reference figure numbers and panel labels in alphanumeric order.

Reviewer #3 (Remarks to the Author):

The study is complete, leveraging a number of techniques, and clearly written. The statistical treatment also appears to be sound, and the illustrations provided appropriately convey the main findings. Overall, I am supportive of this manuscript.

We thank Reviewer 3 for this positive appraisal of our work.

- (1) I find that the manuscript could become stronger by performing correlational analyses, particularly across trials and within subjects. The authors make a strong claim that behavioral and EEG coupling is leading to increase performance, and this is supported in large by effects that are specific to correct trials. But the authors could also show that trials in which coupling is greater, performance is best.

We thank Reviewer 3 for this suggestion. As requested, we have performed comprehensive new single-trial correlational analyses between neural coupling and each key metric – namely: accuracy, RT, MT, curvature, endpoint displacement, gaze distance, behavioural coupling, and task cue amplitude. Indeed, at the single-trial level, univariately, these analyses show that neural coupling is significantly associated with improved behavioural performance on accuracy, RT, endpoint displacement, and behavioural coupling. These new analyses are reported in **Fig. 9** and the section titled “**Neural coupling is uniquely correlated with endpoint accuracy at the single-trial level**”.

- (2) Partial correlations where movement time, reaction time, curvature (etc.) are all controlled for one another while trying to account for accuracy could be interesting too. Likely movement duration and reaction time, for instance, would be correlated, but one can control for these correlations while trying to indicate which if these variables is important in increasing performance.

We thank Review 3 for this suggestion. In response, we have performed single-trial level multiple regressions with the criterion of neural coupling with univariate predictors of accuracy, RT, endpoint displacement, and behavioural coupling. These variables were selected on the basis of being

significant univariate predictors of the neural coupling of joint control at the single-trial level. The multiple regression results show that endpoint displacement uniquely predicted neural coupling. This result is depicted in **Fig. 9i** and reported in the section “**Neural coupling is uniquely correlated with endpoint accuracy at the single-trial level**”.

- (3) I am curious about the LP and HP categorization. Was this done as a whole, for all variables (for instance, RT and MT), or could Participant A be LP for MT and HP for RT? Similarly, was this done on the average of these variables, or on a trial-by-trial basis? The results presented in Figure 2 are not surprising, LP<joint<HP. Much more interesting, in Figure 3 we see that curvature is LP<HP<joint. If the authors could show that this is true even when allowing LP and HP to vary on a trial by trial basis, in effect the authors would have demonstrated that joint behavior can beat the race-model, statistical facilitation. At the moment it is unclear to me whether the sum is truly greater than its parts, as this has to be demonstrated by beating the race-model.

As suggested, we have now focused in more detail on the comparison between LP solo, HP solo, and joint control. In **Fig. 2**, we have identified the HP solo actor at the experiment level. This allows consistency between these metrics, as the HP solo actor on accuracy (correct vs. error) cannot be identified with precision at the single-trial level. As suggested, in the revised trajectory analyses, we have now identified the HP solo actor as the single trial. The results show that performance on curvature was statistically equivalent for joint control and the HP actor. Therefore, for this metric, joint performance was equivalent to statistical facilitation under the race model. We have included this new point in the revised **Discussion**.

- (4) Endpoints become more accurate under the joint condition, and the authors suggest that trajectories become more linear. This is quantified by the appropriateness of a linear fit. I suggest the authors could more directly calculate curvature (angular velocity over radial velocity) to quantify curvature. This measure would also allow to examine whether the strategy of participants change in solo and joint conditions (for instance, up-fronting any needed curvature within a trajectory). Similarly, this analysis can be conducted on every trial, and thus the authors could truly examine the relation between curvature and endpoint

accuracy. At the moment, the authors are only showing that both these measures change during the joint condition, but there is no sense of one of these variables driving the other.

As requested, we have now measured trajectories using curvature and endpoint accuracy as metrics. The results of these new analyses are depicted in **Fig. 3** and reported in the section “**Joint control reduced trajectory curvature and increased endpoint precision for joint control compared with LP solo control**”.

- (5) Minor comments: I would probably choose a different word than “manoeuvre” (used on a number of occasions), as many take this turn to refer to the act of moving around something. In this case there is no obstacle between the cursor and the target.

As requested, we have substituted the word “guide” for “manoeuvre” throughout the revised manuscript.

- (6) On a number of occasions the authors mention that social interaction is needed for cognition. I would suggest tempering this claim, as this is very much an active area of research.

As requested, we have tempered this claim. We have removed this sentence from the **Discussion**. We have reworded this statement in the **Introduction** as follows: “However, increasing evidence suggests that behaviour and cognition may depend on interactions between individuals¹⁴⁻²³.”

- (7) In several figures the authors used a great number of asterisks (*) to denote significance. Personally to me, it is distracting to see a dozen of these asterisk and likely no reader will count how many there are. I would suggest the authors pick of cut-off, for example $p < 0.001$, and denote these with ***

As requested, we have adopted $p < .001$ as the cutoff. We have updated all figures accordingly. We have updated the **Fig. 1** legend as follows: “This and subsequent figures indicate parametric statistical probability values as follows: $p > .050$ NS, $p < .050$ *, $p < .010$ **, $p < .001$ ***.”

Are two heads better than one?

(8) The timing information in Figure 6 was not as clear to me as it was in the other figures.

Please pick a more descriptive label for the x-axis in the ERP figure.

With arrows and labels, we have updated the timeline in **Fig. 6a** and **Fig. 6c** to mark the timeline as beginning at “Action Control Cue Onset” and ending at “Action Control Cue Offset”. Additionally, we have updated the corresponding **Fig. 6** legend to include the following statement: “Depicted data refer to the pre-action trial period beginning with action cue onset and ending with action cue offset.”

References

1. Visco-Comandini, F. *et al.* Do non-human primates cooperate? Evidences of motor coordination during a joint action task in macaque monkeys. *Cortex* **70**, 115–127 (2015).
2. Sebanz, N., Bekkering, H. & Knoblich, G. Joint action: bodies and minds moving together. *Trends Cogn. Sci.* **10**, 70–76 (2006).
3. Marsh, K. L., Richardson, M. J. & Schmidt, R. C. Social connection through joint action and interpersonal coordination. *Top. Cogn. Sci.* **1**, 320–339 (2009).
4. Sebanz, N. & Knoblich, G. Prediction in joint action: what, when, and where. *Top. Cogn. Sci.* **1**, 353–367 (2009).
5. Masumoto, J. & Inui, N. A leader–follower relationship in joint action on a discrete force production task. *Exp. Brain Res.* **232**, 3525–3533 (2014).
6. Miss, F. M. & Burkart, J. M. Corepresentation during joint action in marmoset monkeys (*Callithrix jacchus*). *Psychol. Sci.* **29**, 984–995 (2018).
7. Takagi, A., Ganesh, G., Yoshioka, T., Kawato, M. & Burdet, E. Physically interacting individuals estimate the partner’s goal to enhance their movements. *Nat. Hum. Behav.* **1**, 0054 (2017).
8. Wahn, B., Karlinsky, A., Schmitz, L. & König, P. Let’s move it together: a review of group benefits in joint object control. *Front. Psychol.* **9**, 918 (2018).
9. Astolfi, L. *et al.* Investigating the neural basis of cooperative joint action. An EEG hyperscanning study. in *2014 36th Annual International Conference of the IEEE Engineering in Medicine and Biology Society* 4896–4899 (IEEE, 2014). doi:10.1109/EMBC.2014.6944721.

10. Sinha, N., Maszczyk, T., Zhang Wanxuan, Tan, J. & Dauwels, J. EEG hyperscanning study of inter-brain synchrony during cooperative and competitive interaction. in *2016 IEEE International Conference on Systems, Man, and Cybernetics (SMC)* 004813–004818 (IEEE, 2016). doi:10.1109/SMC.2016.7844990.
11. Bosga, J. & Meulenbroek, R. G. J. Joint-action coordination of redundant force contributions in a virtual lifting task. *Motor Control* **11**, 235–258 (2007).
12. Ferrari-Toniolo, S., Visco-Comandini, F. & Battaglia-Mayer, A. Two brains in action: joint-action coding in the primate frontal cortex. *J. Neurosci.* 1512–1518 (2019).
13. Kourtis, D., Woźniak, M., Sebanz, N. & Knoblich, G. Evidence for we-representations during joint action planning. *Neuropsychologia* **131**, 73–83 (2019).
14. Szpak, A., Nicholls, M. E. R., Thomas, N. A., Laham, S. M. & Loetscher, T. “No man is an island”: effects of interpersonal proximity on spatial attention. *Cogn. Neurosci.* **7**, 45–54 (2016).
15. Saccone, E. J., Szpak, A., Churches, O. & Nicholls, M. E. R. Close interpersonal proximity modulates visuomotor processing of object affordances in shared, social space. *Atten. Percept. Psychophys.* **80**, 54–68 (2018).
16. Dikker, S. *et al.* Brain-to-brain synchrony tracks real-world dynamic group interactions in the classroom. *Curr. Biol.* **27**, 1375–1380 (2017).
17. Rumiati, R. I. & Humphreys, G. W. Cognitive neuroscience goes social. *Cortex* **70**, 1–4 (2015).
18. Adolphs, R. Cognitive neuroscience of human social behaviour. *Nat. Rev. Neurosci.* **4**, 165–178 (2003).

19. Barsalou, L. W. Grounded cognition. *Annu. Rev. Psychol.* **59**, 617–645 (2008).
20. Jiang, J. *et al.* Leader emergence through interpersonal neural synchronization. *Proc. Natl. Acad. Sci. USA* **112**, 4274–4279 (2015).
21. Lieberman, M. D. Social cognitive neuroscience: a review of core processes. *Annu. Rev. Psychol.* **58**, 259–289 (2007).
22. Sebanz, N. & Knoblich, G. Progress in joint-action research. *Curr. Dir. Psychol.* (2021)
doi:10.1177/0963721420984425.
23. Redcay, E. & Schilbach, L. Using second-person neuroscience to elucidate the mechanisms of social interaction. *Nat. Rev. Neurosci.* **20**, 495–505 (2019).

REVIEWERS' COMMENTS:

Reviewer #1 (Remarks to the Author):

I appreciate the authors' responsiveness to my concerns and most of them have been adequately addressed within the limit of the experiment and the data. I have two remaining minor comments, which should be easily addressable:

1. I agree with the authors that there are different instantiations in the literature of "joint action", but I still have reservations with the pervasive use of the word "coordinate" or "coordination" throughout the manuscript. This is because (1) the goal was common between the two partners and (2) under normal circumstances there was very little new learning about the other subject's intention or action. On the contrary, across many fields of research, coordination typically involves forming some models of the partners' minds and actions, e.g., the coordination game literature in economics and psychology. I think that "joint control" (which the authors already use extensively) is a more accurate and less loaded description of the behavioral process being studied here, and that avoiding the use of "coordination" would serve the authors better in conveying the nature of the study.

2. In keeping with the open science movement, I would strongly recommend (to both the editor and the authors) that the data and scripts of this study (at least the behavioral portion which should be of very reasonable size) be shared at a data repository (e.g., OSF), rather than just stating that they are available "upon reasonable request".

Reviewer #2 (Remarks to the Author):

The authors have profoundly revised their original manuscript by following my comments and have also provided novel interesting data. The study is now more solid and convincing. I have further suggestions, mostly related to the way the literature is quoted, which should be updated with more recent and compelling evidence. This is specified below for each section of the ms. together with additional minor issues.

Introduction:

When referring to laboratory investigations demonstrating that "joint actions depend on the abilities to share goal representations, predict each other's actions, and integrate these predictions with one's own actions" (l. 24, p. 2) the authors should refer to the following works by Della Gatta et al, *Cognition*, 2017; Sacheli et al. *Sci Rep* 2018; Novembre et al. *Neuropsychologia*, 2016. In the context of "behavioural cost [of joint actions] relative to the same actions performed individually" (l. 24, p. 2) it should be quoted the study by Satta et al *Neuropsychologia*, 2017, which quantifies this cost across ages in an isometric task very similar to the one adopted in the present study.

Referring to how concurrent brain activity might reveal latent processes associated with task performance (l. 42, p.2), as suggested in the first revision, more than the reviews, the original studies should be quoted, especially those where simultaneous EEG recording has been performed during joint-action execution (see for example the work by Kourtis et al. 2019 and Novembre et al. 2016), even because the reported reviews are old, limited to technological aspects and not updated to the current expanding literature in the field of the neural bases of joint control.

Discussion:

The experimental apparatus is virtually identical to the one used in a set of behavioral studies performed both in monkeys and human subjects (Visco-Comandini et al. Satta et al. 2017) and in a neurophysiological study in monkeys (Ferrari-Toniolo et al. 2019). Surprisingly, this has not been mentioned in the opening of the Discussion. Furthermore, it would be appropriate to explicitly discuss these studies, not only for their similar apparatus which makes the results directly comparable, but also in the view of the similarities of their results with the present findings. In fact, also in Satta et al, 2017 the RTs were found significantly slower for joint control compared to solo actions, for the adult subjects; as for the reduced trajectory curvature interestingly the same result has been documented on monkeys' behavior in Visco-Comandini et al, as well as the

decrease of ICD during joint actions. To provide the reader with a coherent picture of these behavioral invariances across studies, and more importantly across species, it would be desirable to report and discuss these previous observations.

l. 253-54 p. 10 Times as reported in the text are meaningless if behavioral events are not reported in the referred figure, or if the times are not contextualised in the main text with respect to the task phases.

Fig. 6b A color code bar is missing.

Fig. 1 A time arrow in fig. 1b could help to understand the temporal evolution of the task, even because is quite counterintuitive reading the panels from bottom to top.

Reviewer #3 (Remarks to the Author):

I thank the authors for a very thorough revision. Congratulations on a very interesting manuscript.

REVIEWERS' COMMENTS:

Reviewer #1 (Remarks to the Author):

I appreciate the authors' responsiveness to my concerns and most of them have been adequately addressed within the limit of the experiment and the data. I have two remaining minor comments, which should be easily addressable:

1. I agree with the authors that there are different instantiations in the literature of “joint action”, but I still have reservations with the pervasive use of the word “coordinate” or “coordination” throughout the manuscript. This is because (1) the goal was common between the two partners and (2) under normal circumstances there was very little new learning about the other subject's intention or action. On the contrary, across many fields of research, coordination typically involves forming some models of the partners' minds and actions, e.g., the coordination game literature in economics and psychology. I think that “joint control” (which the authors already use extensively) is a more accurate and less loaded description of the behavioral process being studied here, and that avoiding the use of “coordination” would serve the authors better in conveying the nature of the study.

We have raised this perspective as a discussion point so that readers will have an opportunity to draw their own conclusions. We note that this perspective raises empirical and theoretical avenues for investigation, and we have listed some of the more obvious examples.

2. In keeping with the open science movement, I would strongly recommend (to both the editor and the authors) that the data and scripts of this study (at least the behavioral portion which should be of very reasonable size) be shared at a data repository (e.g., OSF), rather than just stating that they are available “upon reasonable request”.

We have done as requested and have made the data and code publicly available through multiple forums (please see **Data availability** and **Code availability** sections).

Reviewer #2 (Remarks to the Author):

The authors have profoundly revised their original manuscript by following my comments and have also provided novel interesting data. The study is now more solid and convincing. I have further suggestions, mostly related to the way the literature is quoted, which should be updated with more recent and compelling evidence. This is specified below for each section of the ms. together with additional minor issues.

Introduction:

When referring to laboratory investigations demonstrating that “joint actions depend on the abilities to share goal representations, predict each other's actions, and integrate these predictions with one's own actions” (l. 24, p. 2) the authors should refer to the following works by Della Gatta et al, Cognition, 2017; Sacheli et al. Sci Rep 2018; Novembre et al. Neuropsychologia, 2016.

We have done as requested.

In the context of “behavioural cost [of joint actions] relative to the same actions performed individually” (l. 24, p. 2) it should be quoted the study by Satta et al Neuropsychologia, 2017, which

quantifies this cost across ages in an isometric task very similar to the one adopted in the present study.

We have done as requested.

Referring to how concurrent brain activity might reveal latent processes associated with task performance (l. 42, p.2), as suggested in the first revision, more than the reviews, the original studies should be quoted, especially those where simultaneous EEG recording has been performed during joint-action execution (see for example the work by Kourtis et al. 2019 and Novembre et al. 2016), even because the reported reviews are old, limited to technological aspects and not updated to the current expanding literature in the field of the neural bases of joint control.

We have added this caveat to the general principle and cited the above-mentioned work.

Discussion:

The experimental apparatus is virtually identical to the one used in a set of behavioral studies performed both in monkeys and human subjects (Visco-Comandini et al. Satta et al. 2017) and in a neurophysiological study in monkeys (Ferrari-Toniolo et al. 2019). Surprisingly, this has not been mentioned in the opening of the Discussion. Furthermore, it would be appropriate to explicitly discuss these studies, not only for their similar apparatus which makes the results directly comparable, but also in the view of the similarities of their results with the present findings. In fact, also in Satta et al, 2017 the RTs were found significantly slower for joint control compared to solo actions, for the adult subjects; as for the reduced trajectory curvature interestingly the same result has been documented on monkeys' behavior in Visco-Comandini et al, as well as the decrease of ICD during joint actions. To provide the reader with a coherent picture of these behavioral invariances across studies, and more importantly across species, it would be desirable to report and discuss these previous observations.

We have now more fully credited this work including discussion of similarities and our comparative contributions.

l. 253-54 p. 10 Times as reported in the text are meaningless if behavioral events are not reported in the referred figure, or if the times are not contextualised in the main text with respect to the task phases.

We have amended this section to refer the reader to **Fig. 1b**, which shows the trial timeline and indicates unambiguously that significant time points for the control cue preceded action.

Fig. 6b A color code bar is missing.

We have added an additional colour bar and have made a note in the legend that two panels share a common color bar.

Fig. 1 A time arrow in fig. 1b could help to understand the temporal evolution of the task, even because is quite counterintuitive reading the panels from bottom to top.

We have added the arrow.

Reviewer #3 (Remarks to the Author):

I thank the authors for a very thorough revision. Congratulations on a very interesting manuscript.

3) Please remove the running title from the top of each page and please remove the formatting of the title headings. In your final version please ensure that only black font is used.

We have done as requested.

4) Please ensure that all author details are included and follow the guidelines of the attached checklist.

We have done as requested.

5) Please ensure that all required statements are included as described in the attached checklist.

We have done as requested.

6) If figure 1a was not entirely generated by the authors, please ensure you have copyright permission to use this figure.

The figure was created by our colleague using copyright free materials. We have now credited our colleague in the figure caption. We have updated the joysticks in **Fig. 5** to use custom art.

7) Please pay close attention to our policy on the sharing of source data (in the checklist) as well as the comment made by reviewer 1 regarding this.

As noted above, we have done as requested.

8) We suggest that you change your title to a shorter, declarative statement. We suggest something along the lines of 'Joint control of visually-guided actions involves concordant changes in behavioural and neural coupling'.

We have made small modifications to your suggestion: **“Joint control of visually guided actions involves concordant increases in behavioural and neural coupling”**.